# SEVtras delineates small extracellular vesicles at droplet resolution from single-cell transcriptomes

Ruiqiao He[1,4], Junjie Zhu[2,4], Peifeng Ji [1] ✉ & Fangqing Zhao [1,2,3] ✉

Small extracellular vesicles (sEVs) are emerging as pivotal players in a wide range of physiological and pathological processes. However, a pressing challenge has been the lack of high-throughput techniques capable of unraveling the intricate heterogeneity of sEVs and decoding the underlying cellular behaviors governing sEV secretion. Here we leverage droplet-based single-cell RNA sequencing (scRNA-seq) and introduce an algorithm, SEVtras, to identify sEV-containing droplets and estimate the sEV secretion activity (ESAI) of individual cells. Through extensive validations on both simulated and real datasets, we demonstrate SEVtras' efficacy in capturing sEV-containing droplets and characterizing the secretion activity of specific cell types. By applying SEVtras to four tumor scRNA-seq datasets, we further illustrate that the ESAI can serve as a potent indicator of tumor progression, particularly in the early stages. With the increasing importance and availability of scRNA-seq datasets, SEVtras holds promise in offering valuable extracellular insights into the cell heterogeneity.

sEVs play important roles in immune responses, viral pathogenicity and cancer progression by mediating cell–cell communication through their bioactive cargoes[1,2]. Many studies have shown that the quantity and composition of sEVs released by a cell can vary substantially across different contexts, depending on factors such as the mode of biogenesis, cell type and physiologic conditions[3,4]. However, conventional bulk RNA sequencing (RNA-seq)-based methods often combine these complex signals, and therefore fall short in representing the true complexity of the sEV population. Recently, several studies have attempted to deconvolve bulk sEV data to identify their tissue of origin and, in some cases, even specific cell types[5–7]. Other studies have sought to characterize the number and cargo of sEVs using microfluidics[8–10]. However, a major limitation of these attempts is their requirement for isolating and purifying sEVs, leading to the loss of valuable information concerning the microenvironment of the original tissue. There is an urgent need for a methodology capable of simultaneously capturing the heterogeneity of both cells and sEVs. Moreover, no existing technique offers

the ability to resolve sEV heterogeneity at a high-throughput scale, or approaches the level of individual sEVs, without imposing additional constraints. Therefore, the development of an effective method that can discern sEVs within droplet-based scRNA-seq datasets, bridging the gap between sEV biology and single-cell transcriptomics, is urgently needed.

To this end, we have developed an algorithm, named sEV-containing droplet identification in scRNA-seq data (SEVtras), to delineate sEV signals at droplet resolution and assess sEV secretion activity across different cell types. Through extensive evaluations using simulated datasets, scRNA-seq datasets of the human mesenchymal stem cell (MSC) and human embryonic kidney 293 cell lines (293F), as well as CITE-seq datasets, we demonstrated that SEVtras can effectively capture sEV-containing droplets within single-cell transcriptomes. Furthermore, by applying scRNA-seq to mixed populations of MSC and 293F cell lines and conducting experiments to enhance sEV secretion, we demonstrated that SEVtras can accurately decipher sEV secretion

[1]Beijing Institutes of Life Science, Chinese Academy of Sciences, Beijing, China. [2]Key Laboratory of Systems Biology, Hangzhou Institute for Advanced Study, University of Chinese Academy of Sciences, Hangzhou, China. [3]University of Chinese Academy of Sciences, Beijing, China. [4]These authors contributed equally: Ruiqiao He, Junjie Zhu. ✉e-mail: jipeifeng@biols.ac.cn; zhfq@biols.ac.cn

patterns across diverse cell types. Our analysis further extended to the characterization of sEV heterogeneity in 15 normal human tissues and colorectal cancer (CRC) tumor tissues, enabling the identification of a subpopulation of migratory-malignant cells with elevated sEV secretion activity. Finally, we highlighted SEVtras' potential use as an effective indicator for early-stage tumor vascular invasion in pancreatic ductal adenocarcinoma (PDAC). In summary, these results underscore the capacity of SEVtras to offer insights into the cell heterogeneity through the study of secreted sEVs.

## Results

### Feasibility assessment of identifying sEV-containing droplets

To identify sEV-containing droplets in scRNA-seq data, the preservation of sEVs during the preprocessing steps of scRNA-seq experiments is a crucial concern. To address this, we assessed sEV concentration before and after the standard scRNA-seq preprocessing steps using three different tissue types. We first observed the presence of a considerable amount of sEVs in these tissues using nanoparticle tracking analysis (Extended Data Fig. 1a,b). We then used NanoLuc-labeled sEVs, which emit luminescence on the introduction of specific substances (Methods). These labeled sEVs were spiked into the samples and subjected to the standard scRNA-seq preprocessing steps (Fig. 1a and Extended Data Fig. 1c–f). We observed a substantial retention of labeled sEVs after undergoing the scRNA-seq preprocessing steps. These findings provide evidence that a considerable proportion of sEVs can be retained and captured during the scRNA-seq process.

Another concern is whether sEVs exhibit distinct gene expression profiles compared to other cellular components. To investigate this, we generated exclusive bulk RNA-seq datasets using MSC and 293F cell lines. For each cell line, we generated three transcriptome datasets that exclusively contained sEVs, cell debris or large EVs, respectively (Methods). To mimic the cell debris generated during scRNA-seq, we subjected cells to mechanical grinding for 4 min (Extended Data Fig. 2). We observed a substantial enrichment of genes in the sEV-exclusive dataset when compared to the exclusive datasets of cell debris and large EVs across both cell lines (Fig. 1b). This enrichment serves as a foundation for distinguishing sEV-specific signals from unrelated cell compartments.

### Overview of SEVtras

For each droplet-based scRNA-seq experiment, the library is composed of cell-containing and cell-free droplets, the latter of which may be further categorized as vacant, cellular debris-containing or sEV-containing droplets. For sEV identification, the key challenge is to separate sEV-containing droplets from cellular debris, as the vacant droplets can easily be filtered by validated barcodes and unique molecular identifier (UMI) counts. To address this challenge, we developed SEVtras to identify sEV-containing droplets based on sEV-specific expression signals and expectation–maximization algorithm (Fig. 1c).

Our strategy starts by building a manually curated sEV-associated RNA gene set from three public sEV-specific gene databases[11–13], referred to as the SEV gene set, which contains 2,017 genes (Supplementary Table 1). Rather than simply performing hypergeometric enrichment of the SEV gene set, which is not tailored for the different biological backgrounds of each dataset, SEVtras uses an expectation–maximization framework to infer the sEV signal score of each droplet to ensure robustness and accuracy in identification (Methods). The logarithmic sEV signal score after convergence of expectation–maximization iterations, referred to as the SEVtras score, is used as a proxy for reliably classifying cell-free droplets into sEVs and debris.

Specifically, we introduce a latent variable $Z$ to encode the sEV signal score within a given droplet. Benefiting from the expectation–maximization algorithm, SEVtras is initialized with the SEV gene set and seeks to converge a representative gene set in a data-driven manner. In this process, SEVtras alternates between an expectation step and a maximization step. During the expectation ($E$) step, each droplet is ranked based on the latent variable $Z$, which is calculated using hypergeometric enrichment analysis of the representative gene set. The subsequent maximization ($M$) step optimizes the gene set by selecting the most representative genes based on the rankings. This iterative procedure continues until the convergence criterion is satisfied. The resulting logarithmic latent variable $Z$ is the SEVtras score used for classification (Methods).

Moreover, to make the SEVtras score comparable across samples, SEVtras aggregates the representative gene set of each sample after convergence by two additional steps: voting and unifying. In the voting step, the frequency of each gene in the converged gene sets is calculated, and only genes occurring in most of these gene sets are included to generate a unified gene set. In the unifying step, the SEVtras scores in all samples are updated with reference to the unified gene set. After sEV droplet identification, further in-depth analyses are performed, including (1) clustering of sEV-containing droplets into different subtypes, (2) sEV signature gene identification and biological function enrichment and (3) estimating sEV secretion acitivity of cells within complex tissue.

### Performance evaluation of SEVtras

To measure the capability of SEVtras, we first sought to generate datasets by simulating cell-free droplets containing either cell debris or sEVs based on mast cells and their sEVs[2] (Methods and Extended Data Fig. 4a). Considering that the fraction of sEVs in cell-free droplets and the overall UMI count of each droplet typically have a strong influence on sEV identification, we generated four gradient parameters for these two variables in the simulation (4 × 4 samples in each dataset) (Supplementary Table 2). In most scenarios, SEVtras exhibited robust performance (AUC > 0.85), indicating that it can tolerate the influence of sEV fraction and UMI count per droplet (Extended Data Fig. 4a).

To further assess the similarity between the sEVs identified by SEVtras and the actual ones, we performed scRNA-seq on experimentally isolated sEVs obtained from MSC and 293F cell lines as ground truth (Fig. 2a). We found a higher expression correlation between the ground truth dataset and the identified sEV-containing droplets by SEVtras, compared to those identified as containing debris (P < 0.001, Mann-Whitney U-test) (Fig. 2b and Extended Data Fig. 4b,c). To strengthen this observation, we performed uniform manifold approximation and projection (UMAP) analysis, which revealed that the distribution of sEV-containing droplets closely overlapped with that of experimentally isolated sEVs, while differing from the distribution of debris (Fig. 2c). Collectively, these findings demonstrate that SEVtras can reliably detect sEV signals within scRNA-seq datasets.

To evaluate the robustness of SEVtras in the presence of complex backgrounds, we conducted spike-in experiments involving cell debris and large EVs (lEVs) of MSC (Methods and Fig. 2d). These two types of spike were separately added to the MSC single-cell suspension, followed by scRNA-seq analysis and SEVtras identification. Transcription profiles of cells were highly overlapped, indicating spike-in did not affect cell state (Fig. 2e). Here, we introduced a new metric, the sEV secretion activity index (ESAI), to quantify the sEV secretion activity of cells. The ESAI is calculated as the ratio of the number of sEV-containing droplets to the number of cells within a given sample:

$$ESAI = \frac{\sum sEV\text{-containing droplets}}{\sum Cell\text{-containing droplets}}$$

We observed no significant change in the ESAI of either spike-in sample compared to the untreated MSC sample (P > 0.05, chi-square test). Moreover, the gene expression profiles of sEVs in both spike-in samples exhibited a high correlation with the untreated sample (Extended Data Fig. 4d) (MSC + debris: Pearson correlation R = 0.999; MSC + large EVs: Pearson correlation R = 0.997). These findings affirm that SEVtras can accurately and specifically identify the signals of sEVs even in the presence of complex backgrounds.

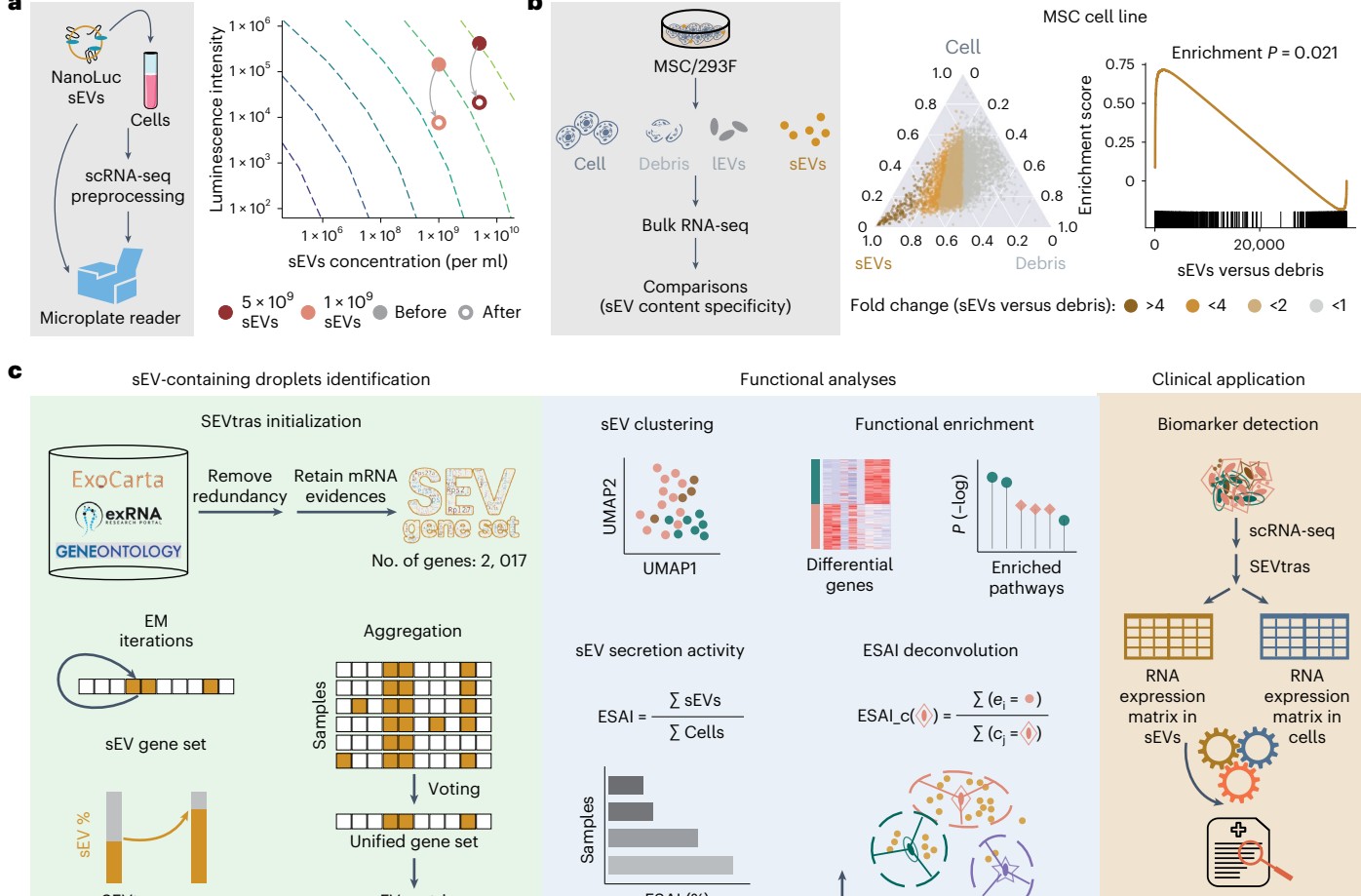

**Fig. 1 | SEVtras learns sEV transcriptome-wide patterns at single-droplet resolution. a**, Changes in luminescence after scRNA-seq preprocessing steps. NanoLuc-labeled sEVs were used to monitor the retention of sEVs during the scRNA-seq procedure. **b**, Transcriptional profile of sEVs is distinct. The left shows the comparisons between sEVs and debris in the MSC cell line. The middle (ternary plot) shows the expression of genes in MSC cell, sEVs and debris. The right (GSEA plot) shows the enrichment score of the SEV gene set in sEVs compared to debris in MSC. $P$ indicates the family-wise error rate (FWER) $P$ values in the GSEA using a hypergeometric test. **c**, Overview of SEVtras. The left panel shows what is initialed by the SEV gene set, SEVtras uses expectation–maximization (EM) iterations to identify sEV-containing droplets from massive cell-free droplets in scRNA-seq data. On achieving convergence through these iterations, SEVtras assigns a classification score to each droplet. To ensure the comparability of SEVtras scores across different samples, the results from each sample are aggregated through a voting and unification process. The middle panel shows the functional analyses for these droplets identified by SEVtras. The right panel shows that SEVtras enables large-scale screening of cell-sEV pairwise signatures in the clinic.

To further assess the specificity of SEVtras in identifying sEV-containing droplets, we stimulated the secretion activity of MSC by introducing monensin sodium salt (MON) to the cultured MSC cells[14] (Fig. 2f). Additionally, we incorporated three times isolated sEVs into the single-cell suspension to further evaluate the assay. After implementing SEVtras, we observed an enhanced ESAI in the two samples compared with untreated MSC cells (Fig. 2g). To further stimulate a low level 'secretion' of sEVs, we treated MSC cell line with MON and simultaneously introduced 1× isolated sEVs (Methods and Extended Data Fig. 4e). As a result, the ESAI of this sample increased by 234% compared to the sample treated with MON alone. Moreover, the increase was about one-third of the gain for the treatment with only 3× isolated sEVs (241%). Based on these findings, we confidently conclude that SEVtras specifically recognizes sEV signals in scRNA-seq sample.

## SEVtras enriches known sEV markers in CITE-seq

To investigate the enrichment of sEV markers in sEV-containing droplets identified by SEVtras, we applied SEVtras to CITE-seq datasets, which allow simultaneous quantification of RNA and surface protein expression within individual cells[15], including two well-documented sEV markers CD63 and CD9. The presence and distribution of these two markers in scRNA-seq data serve as indicators for validating SEVtras. By applying SEVtras, we first validated the enrichment of the SEVtras score in CD63- or CD9-positive droplets, herein sEV marker-positive droplets. We found that these droplets had significantly higher scores than negative droplets ($P < 0.0001$, Mann-Whitney $U$-test) (Fig. 2h and Extended Data Fig. 5a,b). We then measured the proportion of positivity for the two sEV markers in sEV-containing droplets. Notably, SEVtras achieved an approximately twofold enhancement in the fraction of positive droplets compared to a random selection (CD63, 80 versus 44%; CD9, 38 versus 23%) (Fig. 2i). These results collectively indicate a high level of precision and reliability of SEVtras.

In addition to counting the occurrence of marker proteins in droplets, we validated SEVtras using the expression levels of these two proteins. We determined the protein expression in each droplet and found that the abundance of these proteins was significantly higher in sEV-containing droplets than in randomly selected droplets (fold change 1.66) ($P = 0.02$, $t$-test) (Fig. 2j). Similar observations were also found for other sEV-characteristic proteins, such as tetraspanins (for example, CD106) and functional proteins (for example, CD20)

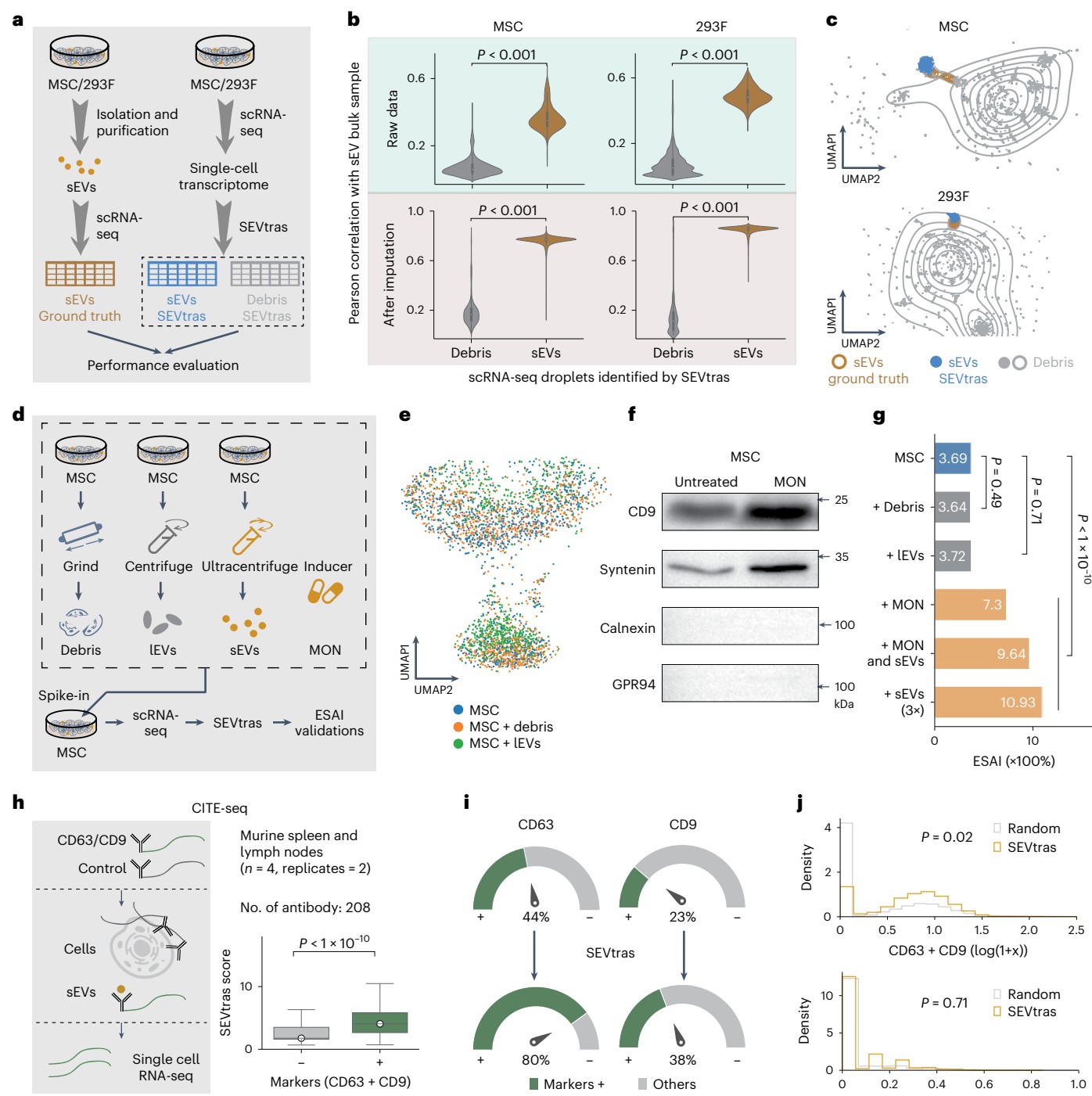

**Fig. 2 | Performance validation of SEVtras. a**, Workflow of ground truth data generation. **b**, sEVs identified by SEVtras were significantly correlated with experimentally isolated sEVs in both MSC and 293F cell lines. For the SEVtras identification data, droplets were classified into two classes: sEV-containing (yellow) and debris-containing (gray) by SEVtras in the boxplot. The top panel shows the Pearson correlation between droplets identified by SEVtras and experimentally isolated sEVs in MSC and 293F cell lines. The bottom shows matrices of debris and sEVs were imputed by MAGIC (Methods). Data are shown as median values with interquartile range. **c**, sEV-containing droplets identified by SEVtras matches the location of ground truth data in UMAP. sEVs SEVtras (blue) and Debris (gray) scatters represent sEV- and debris-containing droplets identified by SEVtras, respectively. Debris (gray) and sEVs ground truth (yellow) lines represent the contour map of the density of debris and experimentally isolated sEVs, respectively. **d**, Schema of the spike-in experiments in MSC cell line. Cell debris and lEVs were introduced as sources of noise in the SEVtras

identification process. Additionally, sEVs and/or a sEV secretion stimulator (MON) was added to assess the specificity of SEVtras. **e**, Cell states in untreated and debris and/or lEVs added MSC samples. **f**, Western blot of sEV marker proteins of untreated and MON-treated samples (n = 2 independent replicates). **g**, ESAI of these spike-in samples with different treatments. P represents the P value in a chi-square test. **h**, The left panel shows a schematic of CITE-seq, which is a method for performing scRNA-seq and gaining information on surface proteins including well-documented sEV markers. The right panel shows the correlation between the SEVtras score and the presence of sEV-specific markers. P represents the P value in a Mann-Whitney U-test. Data are shown as median values with interquartile ranges. **i**, The fraction of CD63- and CD9-positive droplets before and after SEVtras analysis. **j**, The abundance of 'CD63 + CD9' and 'Control IgG' before and after SEVtras processing. P represents the P value in a t-test. The plus sign denotes the summation of the abundance of the two proteins. All statistical tests are two-sided.

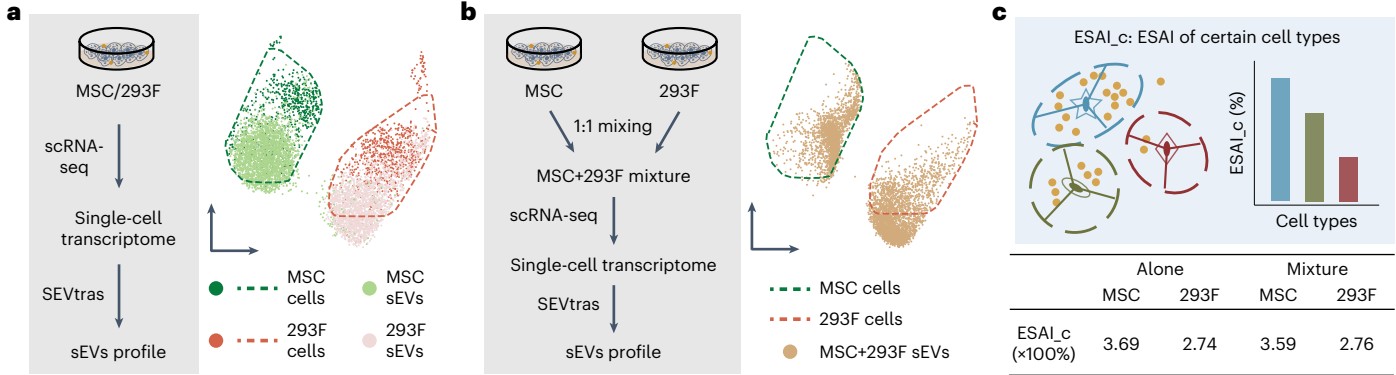

**Fig. 3 | SEVtras unveils sEV secretion activity of certain cell types. a**, The left panel shows the workflow of the scRNA-seq data generation for MSC/293F cell line. The right panel shows the heterogeneity of sEVs in different cell types. Droplets were subsampled to 6,000 for visualization. **b**, The left panel shows the workflow of the scRNA-seq data generation for a mixture of MSC and 293F cells. The right panel shows sEV-containing droplets were highly concordant with sEV-specific clusters of MSC and 293F. Droplets were subsampled to 5,000 for visualization. **c**, The top panel shows a schema of deconvolving ESAI for different cell types. The bottom panel shows that ESAI_c can accurately decipher the sEV secretion activity of different cell types.

(Extended Data Fig. 5c), in contrast to the case for other cell surface proteins (for example, IgG) (Fig. 2j). Moreover, functional enrichment analysis of these sEV-containing droplets highlighted pathways associated with sEV formation and release ($P < 0.05$, hypergeometric test) (Extended Data Fig. 5d). Finally, we evaluated the specificity and precision of SEVtras in the CITE-seq dataset based on marker presence (CD63 and CD9), and these two parameters reached 99.6 and 81.0%, respectively (Extended Data Fig. 5e). The loss (Extended Data Fig. 5f) in precision was mainly due to the extremely low number of genes in the sEV marker-positive droplets caused by the dropout effect during the experiment.

### Deconvoluting sEV secretion activity across various cell types
In our initial application of SEVtras, we focused on investigating the heterogeneity observed in sEVs derived from different cell types. We performed UMAP analysis to compare the transcriptional profiles of sEVs originating from MSC and 293F cell lines (Fig. 3a). Our analysis revealed a distinct separation between the sEVs derived from these two cell lines. However, it should be noted that the sEVs from both cell lines overlapped with their respective original cells. This finding indicates that sEVs originating from diverse cell types possess distinct features, thereby forming the basis for the accurate deconvolution of sEV secretion activity associated with different cell types within the complex microenvironment of tissues.

To further validate the accuracy of SEVtras in an intricacy scenario, we conducted scRNA-seq on a mixed sample containing cells from both MSC and 293F cell lines at a 1:1 ratio (Fig. 3b and Extended Data Fig. 6a,b). The ESAI of this mixed sample (320%) was consistent with the averaged value of ESAI for MSC and 293F (322%). UMAP analysis showed that the sEV-containing droplets from this mixture formed two distinct and heterogeneous clusters. These two clusters of sEVs were highly similar to the clusters observed in the previous analysis of sEVs and original cells (Fig. 3a).

Based on these findings, we were able to use the identified sEV-containing droplets to deconvolve the sEV secretion activity to certain cell type. Subsequently, we deconvolve the ESAI to the cell type level, specifically referred to as ESAI_c, which is determined by dividing the number of sEV-containing droplets of a particular cell type by the total number of cells belonging to that specific cell type $c_i$ (Methods and Fig. 3c).

$$ESAI\_c = \frac{\sum sEV\text{-containing droplets}(Celltype = c_i)}{\sum Cell\text{-containing droplets}(Celltype = c_i)}$$

We found that ESAI_c for MSC and 293F were 359 and 276%, respectively. These values were close to the values obtained for each cell line sample alone (MSC 369%, 293F 274%). After benchmarking with the state-of-art deconvolution methods[6,16], we found ESAI_c can accurately decipher the sEV secretion activity of different cell types (Extended Data Fig. 6c).

### sEV heterogeneity across 15 normal human tissues
To demonstrate the potential of SEVtras for extending single-cell transcriptomics analysis, we applied this algorithm to scRNA-seq dataset of 15 normal human tissues (Fig. 4a and Extended Data Fig. 7a,b)[17]. Overall, a total of 657,407 droplets were obtained after barcode validation and UMI filtration. By leveraging ESAI for each tissue, we found that tissues varied in sEV secretion ability (ESAI = $1.1 \pm 2.0\%$) (Fig. 4b). Notably, the blood sample ranked first based on ESAI, consistent with previous studies[18,19]. Skin also showed a higher ESAI than other tissues, in line with observations that white adipocytes within the dermis possess a high sEV-secreting capacity[20,21]. These results collectively demonstrate that ESAI effectively represents the sEV secretion activity of different tissues and that SEVtras can capture tissue heterogeneity in sEV secretion.

We next focused on delineating the heterogeneity of these identified sEV-containing droplets. After applying the Leiden algorithm with a resolution of 0.5, we found that these droplets were clustered into two subtypes, hereafter called sEV1 and sEV2 (Fig. 4c). We performed differential expression analysis between these two subtypes, and identified 23 differentially expressed genes (DEGs) (Benjamini-Hochberg-corrected $P < 0.05$, Mann-Whitney $U$-test) (Extended Data Fig. 7c). Gene set enrichment analysis (GSEA)[22] of these DEGs revealed that sEV1 was significantly enriched in the mitotic spindle pathway ($P = 0.02$) (Fig. 4c), consistent with the biological functions of Exo-L reported in previous work[23]. By contrast, sEV2 was significantly enriched in the PI3K/Akt/mTOR signaling pathway ($P < 0.01$) (Fig. 4c), which are known as typical sEV functions in cell-cell communication[24]. Thus, SEVtras greatly facilitates in-depth investigation of sEV functional heterogeneity in single-cell transcriptomes.

### High sEV secretion activity in migratory-malignant CRC cells
Mounting evidence have demonstrated the etiological role of sEVs in tumor progression and metastasis[3,4,24]. Precise characterization and tracking sEVs back to their cellular context can have tremendous clinical implications and foster the assessment of tumor progression and prognosis. We applied SEVtras to a CRC scRNA-seq dataset[25] with 27 samples obtained from different tumor locations (Fig. 4d and Extended Data Fig. 7d). In total, we identified 7,709 sEV-containing droplets from

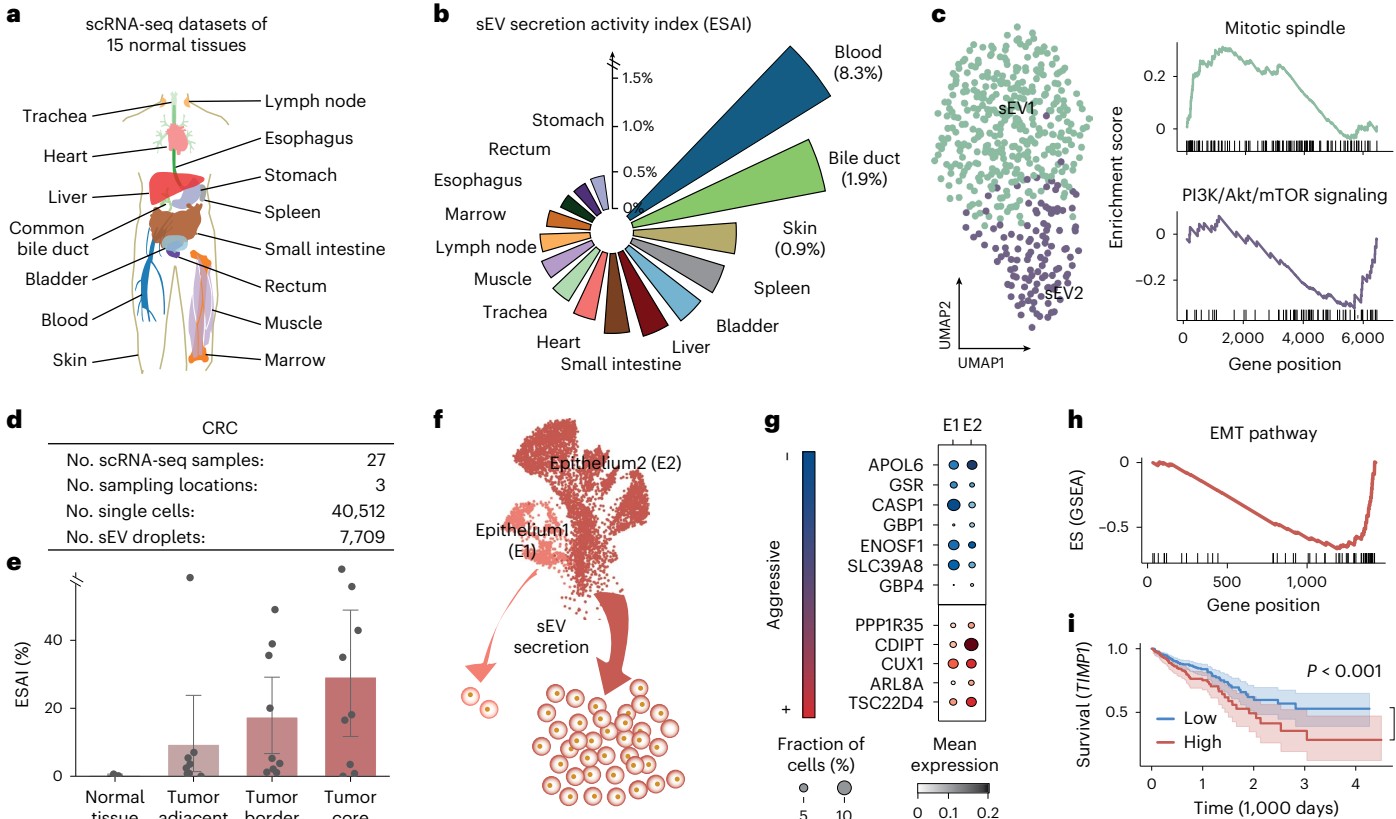

**Fig. 4 | SEVtras deepens single-cell transcriptome analysis. a**, Diagram of the normal human dataset. **b**, Heterogeneity of the ESAI of these normal tissues in this study. **c**, UMAP (left) and GSEA enrichment (right) of the two sEV subtypes. **d**, Description of the CRC dataset. **e**, ESAI at different locations relative to the tumor core (n = 27). The bar represents the mean, and the lower and upper limits in the error bar are the values corresponding to the 2.5th and 97.5th percentiles after 1,000 bootstrap iterations. **f**, Diagram of ESAI_c in the two epithelial subtypes. **g**, Expression of cancer aggression genes in the two epithelial subtypes. **h**, GSEA enrichment of the epithelial–mesenchymal transition (EMT) pathway in the epithelium2 cells. **i**, Survival analysis of *TIMP1* in the TCGA CRC dataset. *P* represents the *P* value in two-sided log-rank test. Data are shown as median ±95% confidence interval.

these scRNA-seq data. The average ESAI in tumor tissues was 18.7%, much higher than that in normal tissues (1.1%) (*P* < 0.01, Mann-Whitney *U*-test), confirming the exceptionally high sEV secretion activity of tumor cells. We then compared the ESAI scores across different sampling sites and found a strong correlation between ESAI and the distance to the core of tumor tissue (Fig. 4e).

Considering that the gut epithelium is a notorious cancerous cell type in CRC[26], we next investigated the sEV secretion activity of different epithelial cell subpopulations. We identified two epithelial subpopulations, epithelium1 and epithelium2, marked by the expression of *FABP1* (Extended Data Fig. 7e,f), in which the low expression level of this gene was closely related to the development of CRC[27]. We found that the ESAI_c of epithelium2 was higher than that of epithelium1 (Fig. 4f and Extended Data Fig. 7g,h). We further focused on cancer aggressiveness-related genes[28] and found that they were indeed enriched in epithelium2 (for example, *TSC22D4* and *CDIPT*) (*P* < 0.05, Mann-Whitney *U*-test) (Fig. 4g and Extended Data Fig. 7i). Subsequent enrichment analysis of DEGs between these two epithelial subtypes revealed that migratory-malignant associated pathways, such as cell migration and epithelial–mesenchymal transition, were strongly enriched in epithelium2 (*P* < 0.05, Mann-Whitney *U*-test) (Fig. 4h). Survival analysis also indicated that a large proportion of epithelium2-enriched genes, for example *TIMP1*, were prognostic markers in CRC[29] (Fig. 4i). Collectively, these results indicate that the aggressiveness of tumor cells associated with sEVs can be evaluated by SEVtras, which may be broadly applicable in cancer progression diagnosis.

**sEV secretion activity links to vascular invasion in PDAC**

To further explore the broad applicability of SEVtras in complex disease contexts, we collected scRNA-seq datasets of four different cancers (Fig. 5a), including PDAC[30], gastric cancer[31], prostate cancer[32] and CRC[25], which comprised 77 samples and more than 220,000 cells (Supplementary Tables 3 and 4). We first calculated the ESAI of these samples and observed that the average ESAI was 10.6 ± 12.3%, corroborating the elevated sEV secretion activity in tumors compared with normal tissues. The ESAI also varied across different patients and studies. In the variance analysis, we found that the most influential factor contributing to this variability was whether the sample originated from a tumor (*P* < 0.01, analysis of variance test). This result was obtained after eliminating the batch factor with a linear regression model.

We next sought to explore the role of sEV secretion activity in tumor progression by matching clinical information to these scRNA-seq datasets. In the PDAC dataset, we found a high positive correlation between vascular invasion and ESAI (*P* = 0.01, Mann-Whitney *U*-test) (Fig. 5b) compared with other factors (Extended Data Fig. 8a). Notably, the ESAI was more distinguishable at stage I (Extended Data Fig. 8b), suggesting that the ESAI may indicate tumor progression at the early stage. In addition, the ESAI was more relevant in different tumor stages when patients were separated by vascular invasion than when they were mixed (Extended Data Fig. 8c,d). Collectively, these results indicate that the ESAI is a promising indicator in the early diagnosis of benign to malignant cancer and may also lead to insights into mechanisms governing tumor progression and metastasis.

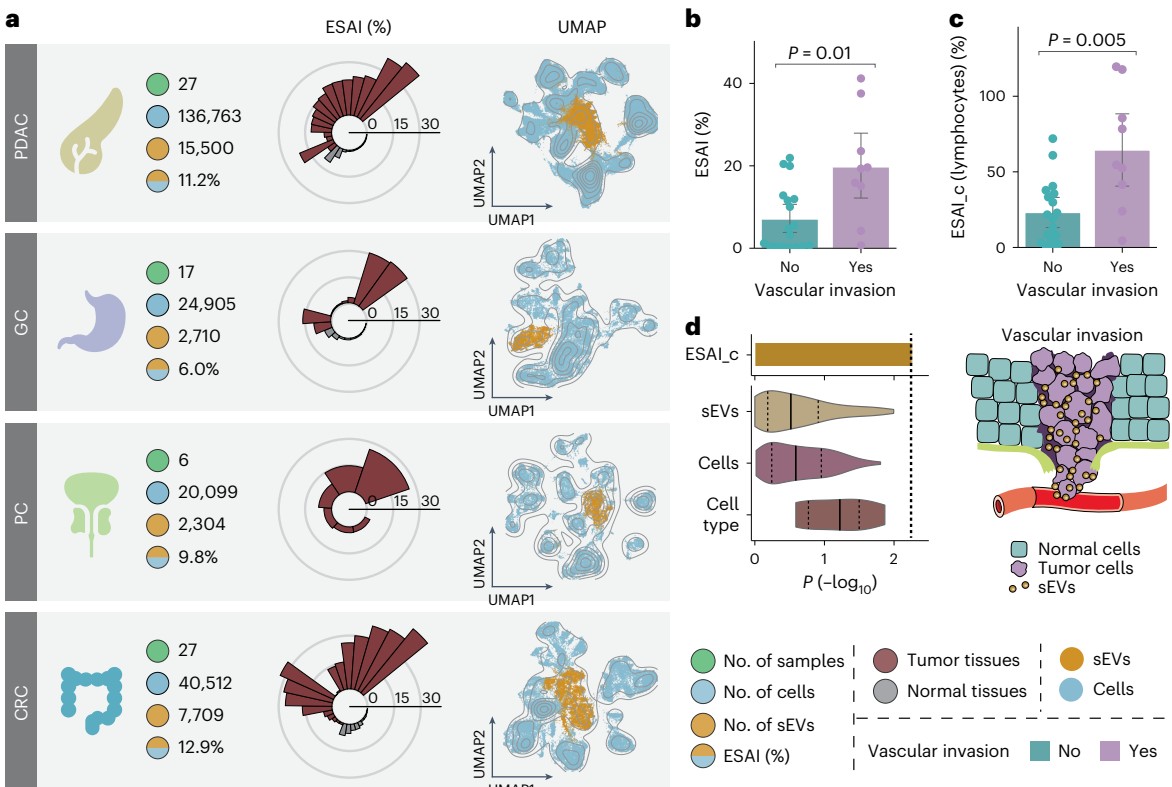

**Fig. 5 | ESAI_c is strongly associated with vascular invasion in early-stage tumors. a**, Schematic of sEV profiling for four cancer types (PDAC, gastric cancer (GC), prostate cancer (PC) and CRC). The left panel shows the meta-information of each dataset. The middle panel shows ESAI varying in the normal and tumor samples. The right panel shows UMAP of identified sEVs and cell types. **b**, ESAI correlates with tumor vascular invasion ($n = 27$). The bar represents the mean, and the lower and upper limits in the error bar are the values corresponding to the 2.5th and 97.5th percentiles after 1,000 bootstrap iterations. **c**, ESAI_c of lymphocytes significantly correlates with vascular invasion ($n = 27$). The bar represents the mean, and the lower and upper limits in the error bar are the values corresponding to the 2.5th and 97.5th percentiles after 1,000 bootstrap iterations. **d**, The left panel shows a comparison of ESAI_c and factors including gene expression and cell type fraction in indicating vascular invasion. ESAI_c is represented by the value of lymphocytes' ESAI_c. The right panel shows a diagram explaining the theoretical basis of ESAI_c in indicating tumor vascular invasion. All statistical tests are two-sided and all $P$ are $P$ values in a Mann-Whitney $U$-test.

To determine the trigger of the elevated sEV secretion activity in vascular invasion, we tracked sEVs back to their original cell types and calculated the ESAI_c. We found that the ESAI_c of lymphocytes was significantly higher in the vascular invasion group than in the group without vascular invasion ($P < 0.005$, Mann-Whitney $U$-test) (Fig. 5c), while no significant difference was observed in the number of lymphocytes ($P = 0.27$, Mann-Whitney $U$-test) (Extended Data Fig. 8e). Moreover, the correlation was more evident at stage I (fold change 3.32) (Extended Data Fig. 8f) than at other stages (fold change 2.25) (Fig. 5c).

To further evaluate the power of ESAI_c in distinguishing tumor vascular invasion, we compared this metric with several other factors related to vascular invasion, including gene expression and cell type fraction. Specifically, we constructed three types of matrix. The first two included the expression profiles of the top 100 highly expressed genes derived from sEVs or cells, and the third contained the fraction of each cell type in scRNA-seq datasets, representing the cellular microenvironment. Then, we calculated the significance of changes within these matrices between the two vascular invasion conditions. Our analysis identified ESAI_c as the most robust indicator, with a significant alteration observed in the ESAI_c of lymphocytes following vascular invasion ($P = 0.005$, Mann-Whitney $U$-test) (Fig. 5d).

## Discussion

We developed SEVtras to provide extracellular insights for scRNA-seq analysis without the need for additional experimental steps. Through analyses of simulated data, as well as real MSC and 293F cell lines and CITE-seq datasets, we have demonstrated the effectiveness of SEVtras

in identifying sEV-containing droplets. Notably, SEVtras exhibited the ability to specifically detect signals from sEVs while remaining resilient to the interference from cell debris and large EVs. To demonstrate the broad applicability of SEVtras, we extended our analysis to scRNA-seq datasets representing 15 normal and four tumor tissue types. Our findings revealed a dramatic increase in ESAI values for migratory epithelial cells in CRC, indicating that their sEVs could potentially transmit signals associated with epithelial–mesenchymal transition.

sEVs actively contribute to tumor migration and invasion, providing valuable insights into the underlying mechanisms of tumor progression and metastasis[4]. Using SEVtras in four distinct cancer datasets, we revealed a significant correlation between sEV secretion activity in lymphocytes and the occurrence of tumor vascular invasion in early-stage PDAC. Notably, this metric outperformed other measurements, including gene expression and cell type fraction, derived from both bulk and single-cell transcriptomic data. These findings underscore the potential of sEV secretion activity as an invaluable complement to single-cell analysis, especially for characterizing tumor samples across the spectrum of benign to malignant states.

We believe that SEVtras will substantially contribute to advancing our understanding of the physiological activities of distinct cell types from the perspective of EVs. One notable aspect of SEVtras is its ability to achieve an exceptional level of resolution in profiling sEVs, allowing for a more comprehensive exploration of the inherent heterogeneity in sEVs originating from diverse tissues. Another key strength of SEVtras lies in its capacity to establish the connection between sEVs and their source cell types, enabling precise quantification of sEV secretion activity specific

to particular cell types within complex tissue microenvironments. Our approach introduces a new dimension for resolving and understanding cellular states by explaining the role of sEVs in mediating cell-to-cell communication. Moreover, SEVtras offers the advantages of high throughput and user-friendliness, seamlessly integrating into various scRNA-seq studies without imposing additional requirements. A limitation of SEVtras is its absence of customized parameters designed to suit diverse tissue types and varying physiological conditions. Further efforts by integrating SEVtras with high-quality tissue-specific gene set and high-resolution detection approach will deepen the field of EV research.

## Online content

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

## Methods

### Overview of the SEVtras method

We first manually curated the sEV-derived messenger RNA gene set, namely the SEV gene set, from three public sEV enriched gene databases, including ExoCarta[11], exRNA Atlas[12] and AmiGO[13] (Supplementary Table 1). We first included genes whose RNA was present in sEVs from these databases. After filtering duplicates, we removed those genes that could not be detected by traditional poly-A tail based scRNA-seq. Finally, genes that met the criteria constituted the SEV gene set.

Based on the SEV gene set, we devised an expectation–maximization algorithm that iteratively learned the sEV signal score for each droplet. Throughout this section, we use $n$ and $g$ subscripts to refer to droplet and gene indices of various vector and matrix variables. $\mathrm{Exp}_{ng}$ is the observed expression level of gene $g$ in cell $n$. $Z_n \in R^z$ is the latent variable indicating the score of sEVs in a certain droplet. $\theta \in R^g$ is the unknown vector denoting the gene set representing sEV signatures. The expectation–maximization process for sEV recognition is described as follows:

In the expectation step, we define $Q(\theta, \theta^i)$ as the expectation of the log-likelihood function of $\theta$ and calculate the latent variable $Z_n$ based on the hypergeometric distribution of the current parameter vector $\theta^i$ using the following equations:

$$Q(\theta, \theta^i) = \sum_{Z_n} P(Z_n | \mathrm{Exp}_{n:}, \theta^i) \log P(\mathrm{Exp}_{n:}, Z_n | \theta)$$

$$P(\mathrm{Exp}_{n:}, Z_n | \theta) \approx \mathrm{Hypergeometric}(k, M, \theta_N, N_n)$$

$$k_n = \sum_{g \in \theta} p_{ng}$$

$$p_{ng} = \begin{cases} 1, & \mathrm{Exp}_{ng} > 0 \\ 0, & \mathrm{Exp}_{ng} = 0 \end{cases}$$

where $k_n$ represents the number of sEV signature genes expressed in droplet $n$, $M$ is the total number of genes in the dataset, $\theta_N$ is the length of $\theta$ and $N_n$ is the number of detected genes in droplet $n$. In this step, SEVtras uses the enrichment of the sEV gene set in the hypergeometric distribution to approach the latent variable $Z_n$ and translates the challenge of identifying sEV droplets into finding a more representative sEV gene set $\theta$.

In the maximization step, the parameter $\theta$ updates by maximizing the expected value of the bound using the following equations:

$$\theta^{i+1} = \underset{\theta}{\mathrm{argmax}}\, Q(\theta, \theta^{i+1})$$

$$\approx \underset{\theta}{\mathrm{argmax}} \prod_g a \frac{\mathrm{cov}(R(\mathrm{Exp}_{:g}), R(Z_{:}))}{\sigma_{R(\mathrm{Exp}_{:g})} \sigma_{R(Z)}}$$

where $a$ is the factor controlling the length of $\theta$ with a default value of 10, $\mathrm{cov}(A, B)$ is the covariance of the two variables ($A$ and $B$), $R(A)$ is the rank of the variable and $\sigma$ is the standard deviation. Here, SEVtras leverages the correlation metric between gene expression and latent variable $Z_n$ and updates the elements in the gene set $\theta$ by seeking the genes most relevant to the latent variable.

At the beginning of the expectation–maximization algorithm, the model parameter $\theta$ is initialized by the SEV gene set. Then, the expectation and maximization steps are executed iteratively until parameter $\theta$ converges. The latent variable $Z_n$ in the final iteration is referred to as the SEVtras score in the given sample.

If there are multiple samples in a dataset, SEVtras first performs expectation–maximization iterations for each sample. After convergence, SEVtras integrates the convergent representative sEV gene set and selects the most frequently occurring gene as the unified gene set by voting. Finally, the SEVtras score is updated based on the unified gene set.

### Simulations

To mimic the transcriptome profile of cell-free droplets, we customized a simulation method based on the zero-inflated negative binomial (ZINB) distributions. Regarding cell-free droplet data simulation, the most crucial step was to control the range of total UMIs following an extremely sparse gene distribution. Therefore, we allowed the number of total UMIs in each droplet to follow a Poisson distribution using the following equations:

$$p_n \approx \mathrm{Poisson}(\mathrm{UMI})$$

$$p_n = \sum_g \mathrm{Exp}_{ng}$$

where UMI represents the expected total UMIs in one droplet and $p_n$ is the UMI count generated in specific droplet $n$. The expression of each gene ($\mathrm{Exp}_{ng}$) is subjected to a ZINB distribution similar to the conventional single-cell simulation method as follows:

$$\mathrm{Exp}_{ng} \approx \mathrm{ZINB}(a, m_{ng}, p_n)$$

where $a$ is the dropout rate and $m_{ng}$ is a gene-specific parameter reflecting the expression level in the droplet. In this work, we generated $m_{ng}$ for sEVs and debris based on the expression of gene $g$ in the bulk transcriptome data from murine mast cells and their sEVs[2].

### ESAI calculation

ESAI at the sample and/or tissue level is defined as the number of sEV-containing droplets divided by the number of cell-containing droplets, defined as follows:

$$\mathrm{ESAI} = \frac{\sum \text{sEV-containing droplets}}{\sum \text{Cell-containing droplets}}$$

where sEV-containing droplets means sEV-containing droplets identified by SEVtras within a given sample, and Cell-containing droplets means cellular droplets captured by scRNA-seq within a given sample.

### sEV secretion activity deconvolution (ESAI_c)

To calculate the sEV secretion activity for different cell type (ESAI_c), we deconvolved sEV-containing droplets into secreted cell types based on two assumptions: (1) the transcriptional profile of a given sEV is more similar to its cell of origin[6,16] and (2) sEV release is affected by the biogenesis capacity of the original cell type.

Specifically, we first used the gene expression similarity to measure the extent to which a given sEV can belong to certain cell types (assumption 1):

$$\mathrm{Similarity}_{ci} = \sum_{j \in \mathrm{Cell}N} [\mathrm{Celltype}(j) = ci]$$

where Cell$N$ refers to the $N$ (default 10) nearest neighboring cells of the sEV droplet in the principal component analysis (PCA) coordinates. We applied 'scipy.spatial.cKDTree' for quick nearest-neighbor lookup. Celltype($j$) represents the cell type of cell $j$, and [·] represents the number of cases satisfying the criterion. We set a limitation as 2 in the Similarity$_{ci}$, if a given sEV is not similar to any cell types, we do not consider it in the later calculation.

The sEV biogenesis capacity represents the threshold for determining whether a given sEV belongs to a certain cell type or not (assumption 2):

$$\mathrm{Biogenesis}_{ci} = \mathrm{GSEA}(\mathrm{Exp}_{n_{ci}g}, \mathrm{sEVGO})$$

$$\mathrm{Source\_Celltype} = \underset{ci}{\mathrm{argmax}}(\mathrm{Similarity}_{ci} + \mathrm{Biogenesis}_{ci})$$

GSEA denotes the GSEA function[22] and sEVGO is the sEV biogenesis gene set from the Molecular Signatures Database[33].

sEV secretion activity at the cell type level (ESAI_c) is the number of sEV-containing droplets secreted by a certain cell type divided the number of cell-containing droplets for a certain cell type (ci), defined as follows:

$$ESAI\_c = \frac{\sum sEV\text{-containing droplets(Celltype = ci)}}{\sum Cell\text{-containing droplets(Celltype = ci)}}$$

where the Celltype in sEV-containing droplets represents the Source_Celltype deconvolved previously.

**Tissue sEV separation and nanoparticle tracking analysis**

sEVs were separated from tissue using the protocol established previously[34], with minor modifications. The dissociation mixture was based on the Miltenyi Dissociation Kit (Miltenyi Biotec, catalog no. 130-095-929). Before starting, enzymes H, R and A were resuspended according to the manufacturer's instructions. Dissociation mix containing 2.2 ml RPMI, 100 µl enzyme H, 50 µl enzyme R and 12.5 µl enzyme A was prepared immediately before use. A small (roughly 200 mg) piece of tissue was weighed and briefly sliced on dry ice and then incubated in the dissociation mixture for 10-15 min at 37 °C. The dissociated tissue filtered through a 70 µm filter gently for twice to remove residual tissues. Then suspension was spun at 300$g$ for 10 min at 4 °C, and supernatant was transferred to a fresh tube and spun at 2,000$g$ for 10 min at 4 °C and filtered with a 0.2 µm filter (FPE-234-000, BIOFIL). This filtered medium was then ultracentrifuged at 100,000$g$ at 4 °C for 4 h, the pellet was then suspended in PBS and re-ultracentrifuged at 100,000$g$ for 20 min. Finally, the pellet was resuspended in 50 µl of PBS. For nanoparticle tracking analysis, isolated sEVs were measured using a NanoSight NS300 (Malvern) equipped with a high sensitivity sCMOS camera. For each acquisition, a delay of 90 s followed by three captures of 30 s each was used. The averaged value of the three captures for each biological replicate was used to determine the nanoparticle concentration and the size distribution.

**Performance evaluation in the MSC and 293F cell line**

Human umbilical cord MSCs were provided by Jinan Wanquan Biotechnology and cultured in basic medium (Dakewe) supplemented with 5% UltraGRO-PURE serum substitutes (Helios BioScience) and 1% penicillin-streptomycin at 37 °C in 5% $CO_2$. Human embryonic kidney cell 293F was from QuaCell Biotechnology and cultured in shake flasks in SMM 293-TII serum-free medium (Sino Biological) and 1% penicillin-streptomycin at 37 °C in 5% $CO_2$. Cultured cells were washed three times in PBS and subsequently allowed to secrete sEVs for 48 h. sEVs were isolated by sequential differential centrifugation from cultured cell medium. The medium was collected from the cell cultures and spun at 800$g$ for 5 min, followed by 2,000$g$ for 10 min and filtered with a 0.2 µm filter (FPE-234-000, BIOFIL). This filtered medium was then ultracentrifuged at 100,000$g$ at 4 °C for 4 h, the pellet was then suspended in PBS, and re-ultracentrifuged at 100,000$g$ for 20 min. Finally, the pellet was resuspended in 50 µl of PBS. Isolated sEVs were stored at −80 °C for downstream analyses.

To examine the purity of the isolated sEVs, nanoparticle tracking analysis and western blotting were used. For nanoparticle tracking analysis, sEVs were measured using a NanoSight NS300 (Malvern) equipped with a high sensitivity sCMOS camera. For each acquisition, a delay of 90 s followed by three captures of 30 s each was used. The averaged value of the three captures for each biological replicate was used to determine the nanoparticle concentration and the size distribution. For western blotting, equal amounts of total protein extracted from MSC or 293F cells and sEVs were subjected to separation by SDS-PAGE and further analyzed. Antibodies against

CD9 (Proteintech, diluted 1:1,500), Syntenin (Proteintech, diluted 1:1,000), Calnexin (Easybio, diluted 1:8,000) and GPR94 (Proteintech, diluted 1:1,000) were used to identify sEV and cellular protein markers according to the manufacturer's instructions. HRP-conjugated goat antibody (Easybio, diluted 1:5,000) was used as the secondary antibody.

Total RNA was extracted from the isolated sEVs using TRIzol (Invitrogen). The library was prepared following the TruSeq protocol (Illumina) and sequenced on an Illumina Next500 sequencer. For the bulk transcriptome analysis, raw reads were cleaned using Trim Galore v.0.6.7 (ref. [35]) and aligned to the GRCh38 human reference genome using STAR v.2.6.1a (ref. [36]) and RSEM v.1.2.25 (ref. [37]) to quantify the transcripts and DESeq2 v.1.26.0 (ref. [38]) was used for sample normalization.

**Generation of cell debris and large EVs**

To generate cell debris, cell suspension was ground with stainless steel beads and unbroken cells were removed by centrifugation at 300$g$ for 5 min at 4 °C. To obtain large EVs, cell suspension was centrifuged at 300$g$ for 5 min to remove cells and large debris, followed by 2,000$g$ for 20 min at 4 °C. The pellet from 2,000$g$ centrifugation was resuspended with PBS and regarded as large EVs[39].

**NanoLuc-labeled sEVs preparation**

Sequence of NanoLuc has been fused with EV sorting motif MysPalm[40] and synthesized. For plasmid construction, MysPalm-NanoLuc was inserted between AflII and AgeI into a pcDNA3.4 vector. The 293F cells were suspended and cultured with serum-free medium (Sino Biological) at 37 °C, 5% $CO_2$. Cells of $2 \times 10^6$ cells per ml and 95% viability were transfected with plasmids using polyethyleneimine transfection reagent (catalog no. 24765-1, Polyscience) for 72 h. After transfection, the cell culture supernatant was collected for sEV isolation by sequential differential ultracentrifuge. The supernatant was spun at 800$g$ for 5 min, followed by 2,000$g$ for 10 min and filtered with a 0.2 µm filter (FPE-234-000, BIOFIL). This filtered medium was then ultracentrifuged at 100,000$g$ at 4 °C for 4 h, the pellet was then suspended in PBS and re-ultracentrifuged at 100,000$g$ for 20 min. Finally, the isolated NanoLuc-labeled sEVs were stored at −80 °C for further analyses.

NanoLuc luminescence emission was measured using Nano-Glo Luciferase Assay (Promega) according to the manufacturer's instructions and read by a multimode microplate reader SpectraMax i3 (Molecular Devices) at 460 nm.

To verify that NanoLuc protein was specifically loaded in engineered sEVs, proteinase K (Qiagen) or Triton X-100 (Thermo Fisher) were used to treat sEVs. A final concentration of 0.1% Triton X-100 was used to permeabilize the membrane that facilitates the digestion of intracellular proteins. A total of four experimental conditions (only NanoLuc-labeled sEVs, NanoLuc-labeled sEVs treated with Triton X-100 only, NanoLuc-labeled sEVs treated with proteinase K only, NanoLuc-labeled sEVs treated with proteinase K + Triton X-100) were set up with three replicates for each condition. After 3 h of incubation at 37 °C, the luminescence of each treatment was monitored using the Nano-Glo Luciferase Assay (Promega) according to the manufacturer's protocol.

**scRNA-seq processing**

The cell samples for scRNA-seq were prepared according to the protocol recommended by 10X Genomics with minor modifications. MSCs and 293F cells were both cultured in serum-free medium following the manufacturer's instructions and cell culture reaching 80% confluency was used for further process. After removing culture medium and washing once with PBS, MSCs were digested with 0.25% Trypsin-EDTA solution (Gibco) at 37 °C for 2 min and the digestion was stopped by adding new culture medium. The cells were then centrifuged at 250$g$

for 5 min, resuspended with culture medium and filtered through a 40 μm cell strainer. The cell concentration of 293F and filtered MSC suspension were determined and diluted between $6 \times 10^5$ and $1 \times 10^6$ cells per ml. Cells was pelleted from 1.5 ml of cell suspension at $150g$ for 3 min and washed with 1× PBS containing 0.04% BSA. Then repeat the centrifuge and wash steps for a total of three washes. The cells were resuspended with 1× PBS containing 0.04% BSA and filtered through a 40 μm cell strainer to a concentration of $7 \times 10^5$ to $1 \times 10^6$ cells per ml and 7,000 cells were used for each sample. The MSC and 293F mixed sample was prepared using 3,500 MSCs and 3,500 293F cells to prepare. Libraries were constructed following standard 10× Chromium Single Cell Gene Expression Protocol (v.3.1) and sequenced on a NovaSeq 6000 platform (Illumina).

**sEV secretion activity stimulation**
MON (MedChemExpress) was dissolved in dimethylsulfoxide to a stock solution of 5 mM and diluted with cell culture medium to a final concentration of 5 μM. The resulting medium was used to treat the MSC cells for 18 h before collecting the culture medium for sEV isolation or cells for scRNA-seq.

**scRNA-seq data processing**
Raw reads were processed by Cell Ranger v.5.0.0 with default parameters[41]. Samples with sequencing saturation less than 0.5 were discarded for downstream analysis. The raw expression matrix of Cell Ranger outputs was inputted into SEVtras for sEV-containing droplets identification. The matrix and downstream analysis for sEVs was performed using the Scanpy v.1.8.2 package[42]. The expression matrix was normalized with 'sc.pp.normalize_total' with a target sum in each droplet as 100. Data imputation was performed by MAGIC v.3.0.0 with default parameters[43]. Then, we identified highly variable genes with 'sc.pp.highly_variable_genes'. PCA was performed on the scaled variable gene expression matrix using 'sc.pp.pca'. Using the first 50 principal components, we constructed a shared nearest-neighbor graph for cells with 'sc.pp.neighbors' and clustered cells using 'sc.tl.leiden'. BBKNN v.1.5.1 was used for batch effect adjustment with default parameters[44]. Finally, we applied the UMAP algorithm using the 'sc.tl.umap' function to visualize cells in low dimensions. The DEGs between clusters were identified by 'sc.tl.rank_genes_groups'.

**Statistical analysis**
Python v.3.8 with numpy v.1.20.3 and pandas v.1.2.4 was used for data analysis, and matplotlib v.3.4.2 and seaborn v.0.11.0 for visualization. All $P$ values resulting from multiple hypothesis testing in all analyses were adjusted with the Benjamini-Hochberg false discovery rate. The adjusted $P$ value is referred to as the Benjamini-Hochberg-corrected $P$ value in the main text. $n$ typically indicates biologically independent experiments.

**Reporting summary**
Further information on research design is available in the Nature Portfolio Reporting Summary linked to this article.

## Data availability
The bulk RNA-seq and scRNA-seq data for MSC and 293F cells were deposited at the National Genomics Data Center with accession number PRJCA017291 (https://ngdc.cncb.ac.cn/gsa-human/browse/HRA004708). The CITE-seq data[15] and scRNA-seq data for 15 normal tissues[17] and prostate cancer[32] were downloaded from the National Center for Biotechnology Information Gene Expression Omnibus (GSE150599, GSE159929 and GSE137829, respectively). The scRNA-seq data for CRC[25] were accessed in ArrayExpress under the accession number E-MTAB-8410. The scRNA-seq data for PDAC[30] were accessed from Genome Sequence Archive under the accession number CRA001160. The scRNA-seq data for gastric cancer[31] were accessed at https://dna-discovery.stanford.edu/research/datasets/. Source data are provided with this paper.

## Code availability
Source code of SEVtras is freely available at https://github.com/bioinfo-biols/SEVtras.

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

## Acknowledgements
This work was supported by grants from the National Natural Science Foundation of China (grant nos. 32025009, 32130020 and 32071463), National Key R&D Project (grant nos. 2021YFA1300500, 2021YFA1301000 and 2021YFA1302000) and Natural Science Foundation of Beijing, China (grant no. Z230007). We thank D. Li and H. Wang for providing MSC cell line resources.

## Author contributions
F.Z. conceived the study. R.H. designed the algorithm. R.H., J.Z. and P.J. analyzed the data and performed experiments. R.H., P.J. and F.Z. wrote the paper.

## Competing interests
The authors declare no competing interests.

## Additional information
**Extended data** is available for this paper at https://doi.org/10.1038/s41592-023-02117-1.

**Correspondence and requests for materials** should be addressed to Peifeng Ji or Fangqing Zhao.

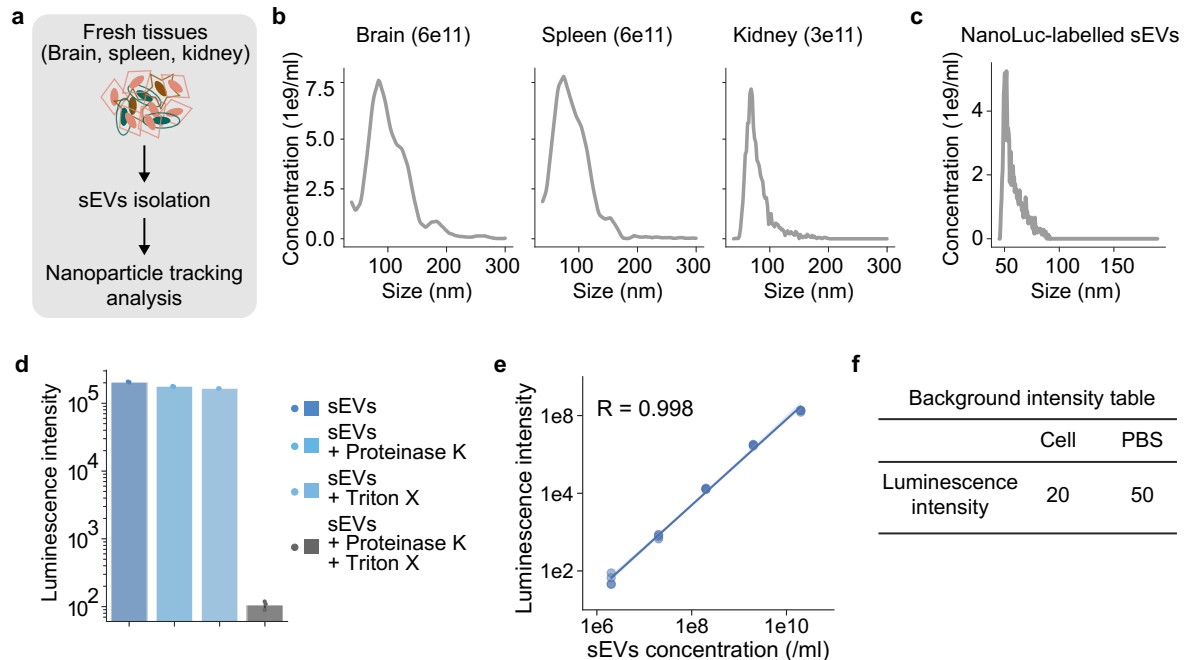

**Extended Data Fig. 1 | Presence of sEVs in the scRNA-seq samples. a**, Schema of nanoparticle tracking analysis (NTA) for fresh tissues. **b**, Size distribution of sEVs in the three types of mouse tissues. From left to right: brain, spleen and kidney. **c**, NTA for NanoLuc-labeled sEVs. **d**, Luminescence intensity of NanoLuc-labeled sEVs treated with Triton X-100 and Proteinase K ($n = 3$ independent experiments for each condition) (see Methods). **e**, Luminescence intensity of NanoLuc-labelled sEVs at variable concentration. **f**, Luminescence intensity of background. The luminescence was measured by microplate reader (see Methods).

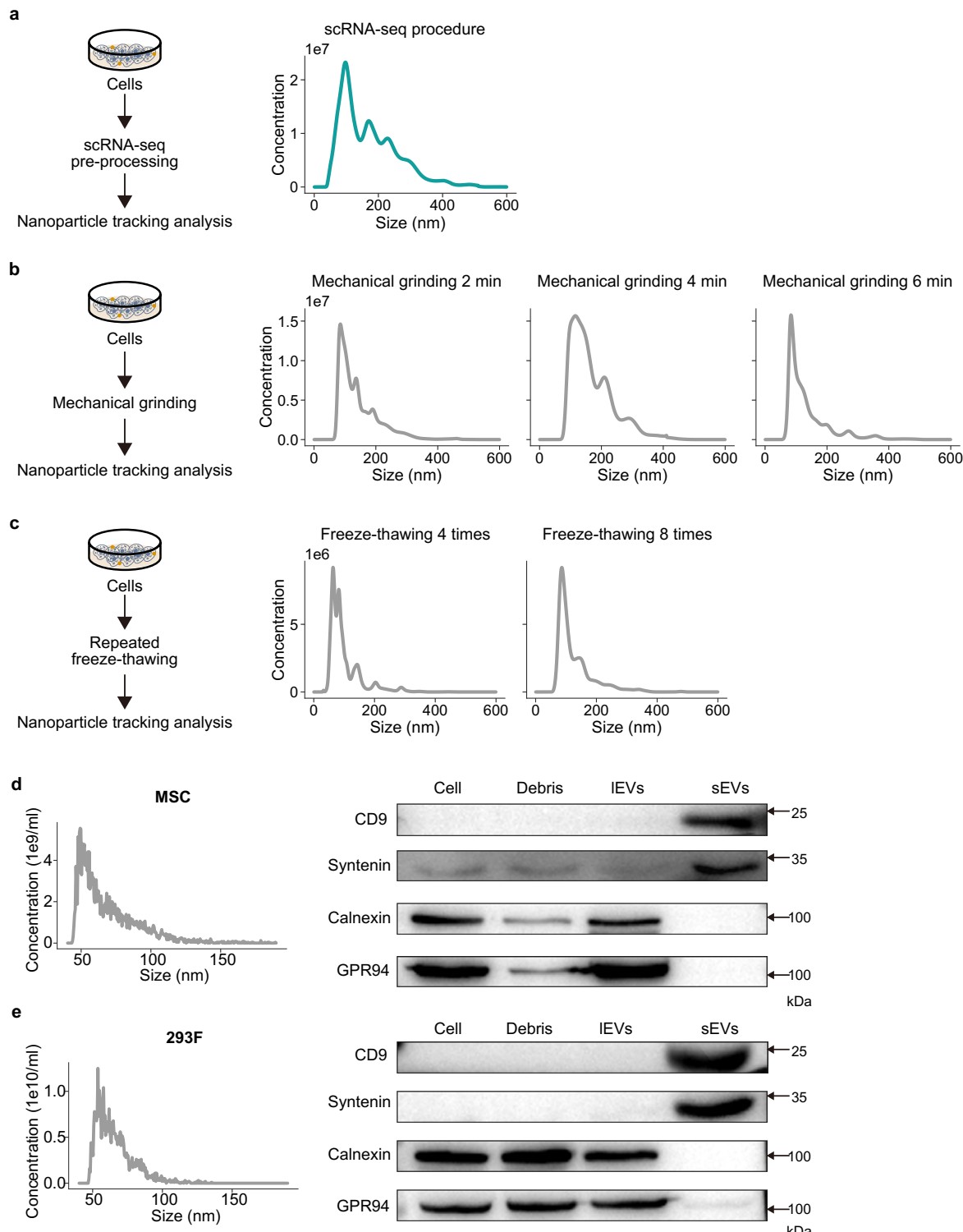

**Extended Data Fig. 2 | Debris generation and sEVs characterization.**
**a**, Nanoparticle tracking analysis of the cells after scRNA-seq pre-processing procedures. **b**, Nanoparticle tracking analysis of the cells after mechanical grinding for different duration. **c**, Nanoparticle tracking analysis of the cells after repeated freeze-thawing at different times. **d**, Left panel: nanoparticle tracking analysis of the MSC sEVs. Right panel: western blot of sEV-positive proteins (CD9 and Syntenin) and sEV-negative proteins (Calnexin and GPR94) in MSC cell, debris, larger EVs (lEVs) and sEVs. $n = 2$ independent replicates. **e**, Left panel: nanoparticle tracking analysis of the 293F sEVs. Right panel: western blot of sEV-positive proteins (CD9 and Syntenin) and sEV-negative proteins (Calnexin and GPR94) in 293F cell, debris, larger EVs (lEVs) and sEVs. $n = 2$ independent replicates.

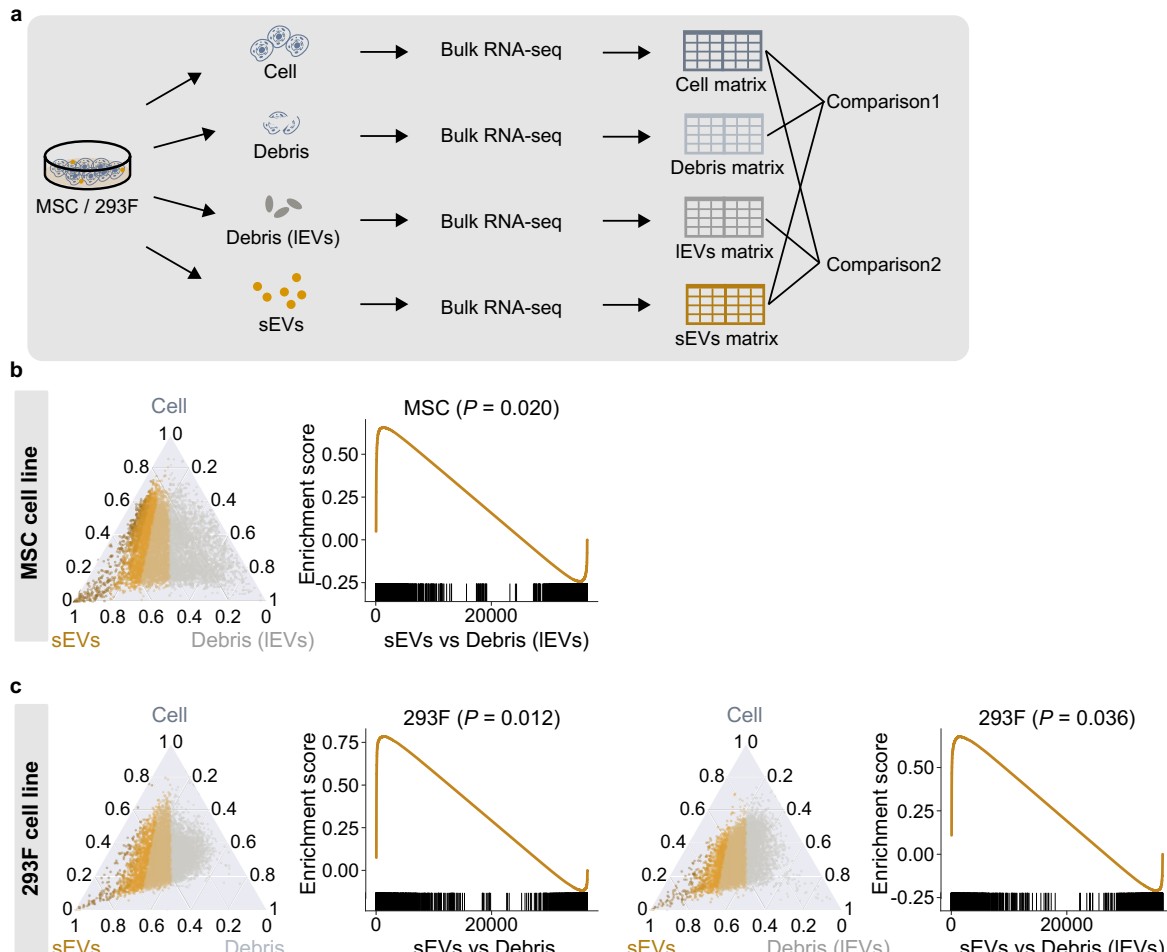

**Extended Data Fig. 3 | Transcriptional profile of sEVs is distinct with that of cell debris and large EVs in both MSC and 293F cell lines. a**, Experimental design of generating exclusive transcriptional profiles of debris, sEVs and large EVs (lEVs). Debris, sEVs and lEVs are isolated from cell cultures. In particular, for cell debris, we mechanically ground MSC/293F cells with four minutes (Extended Data Fig. 2a–c). For lEVs, we centrifuged MSC/293F cell medium at 2,000 × g and collected pellets (see Methods). sEVs is separated by ultracentrifugation (see Methods). **b-c**, Comparisons between sEVs and debris or debris (MVs) in MSC (**b**) and 293F (**c**) cell lines. Left (ternary plot): the expression of each gene in cell, sEVs and debris or lEVs. Color depicts the fold change between sEVs and debris or lEVs. Right (GSEA plot): enrichment score of the SEV-gene set in sEVs compared to debris or lEVs. '*P*' means FWER *P* values in the GSEA using a hypergeometric test. Also see Fig. 1b.

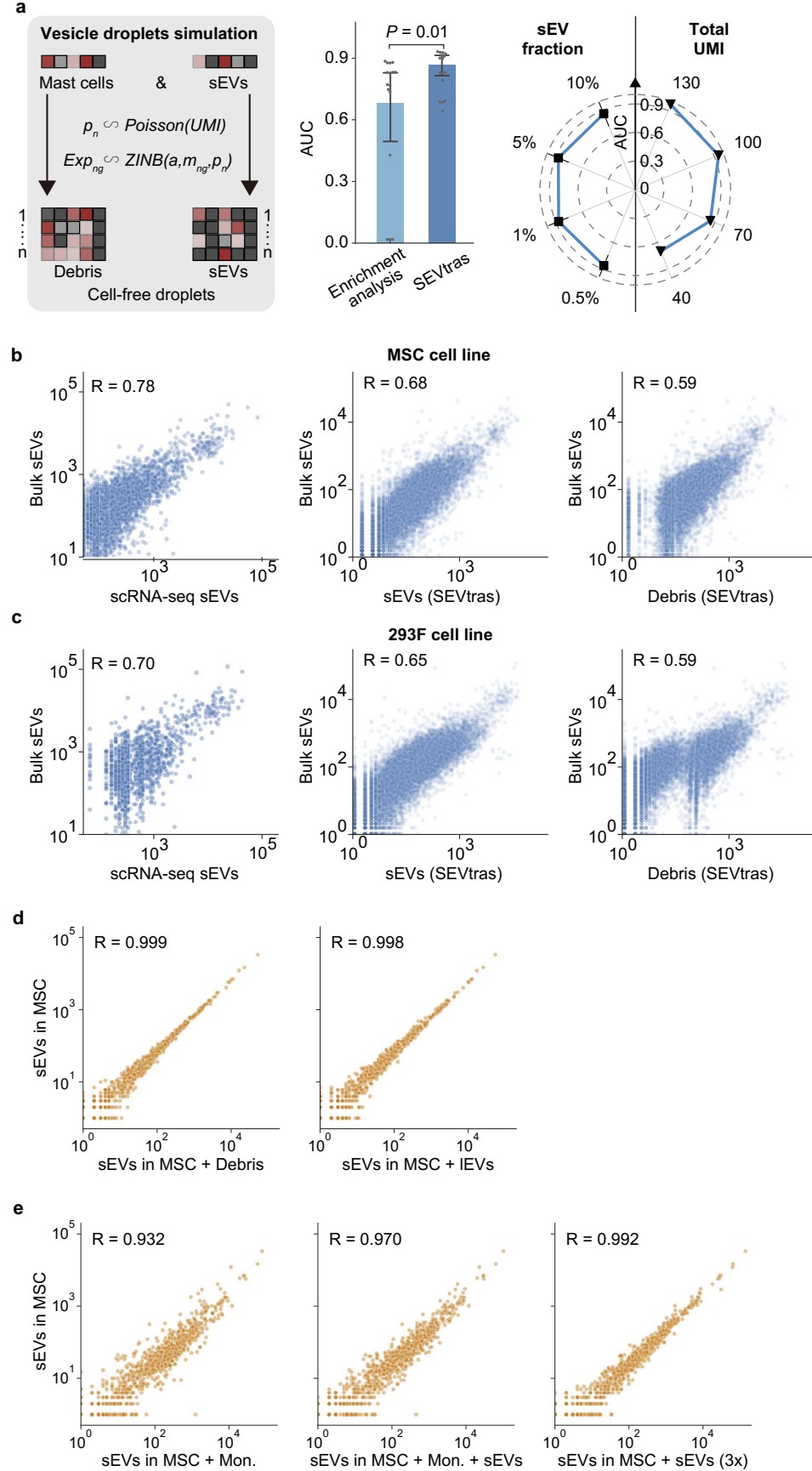

**Extended Data Fig. 4 | See next page for caption.**

**Extended Data Fig. 4 | Performance evaluation for SEVtras. a**, Evaluations by simulated data. Left panel: outline of cell-free droplet data simulation. Middle panel: AUC of SEVtras compared with the method of enrichment ($n = 16$ independent experiments). '$P$' represents the $P$ value in two-sided T test. The bar represents the mean, and the lower and upper limits in the error bar are the values corresponding to the 2.5th and 97.5th percentiles after 1,000 bootstrap iterations. Right panel: AUC estimation of the influence of two confounders (fraction of sEVs and total UMI counts) on SEVtras. **b**, Transcriptional profile correlation among experimentally isolated sEVs, sEVs and debris identified by SEVtras in MSC cell line. **c**, Transcriptional profile correlation among experimentally isolated sEVs, sEVs and debris identified by SEVtras in 293F cell line. **d**, Gene expression of sEVs was highly correlated in untreated and debris/lEVs added MSC samples. 'lEVs' means large EVs. **e**, Gene expression of sEVs was highly correlated in untreated and MON treated/sEVs added MSC samples. 'MON' means monensin sodium salt.

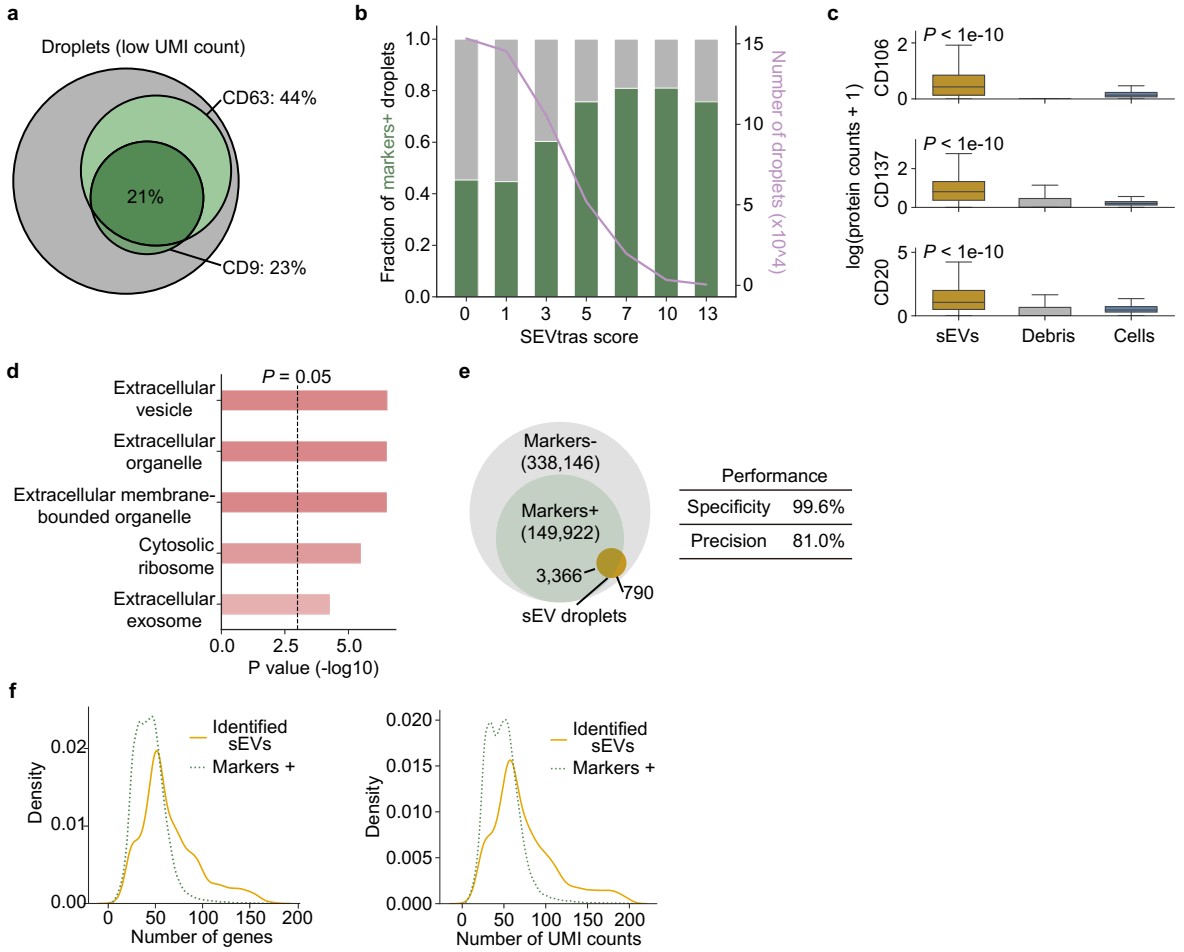

**Extended Data Fig. 5 | SEVtras enriches well-documented sEV markers in CITE-seq. a**, Presence of CD63 and CD9 in the CITE-seq data. **b**, The fraction of sEV marker positive droplets under different thresholds. **c**, The abundance of CD20, CD137 and CD106 in droplets ($n$ =186,434). '$P$' represents the $P$ value in two-sided Mann-Whitney U test. Data are shown as median values with interquartile range. **d**, GO enriched terms of sEV droplets identified from the CITE-seq data. '$P$' represents the $P$ value in GO enrichment using a hypergeometric test. **e**, Performance validation of SEVtras based on the two sEV-specific markers. **f**, Distribution of gene number (left) and total UMI count (right) between droplets identified by SEVtras and droplets positive for sEV markers.

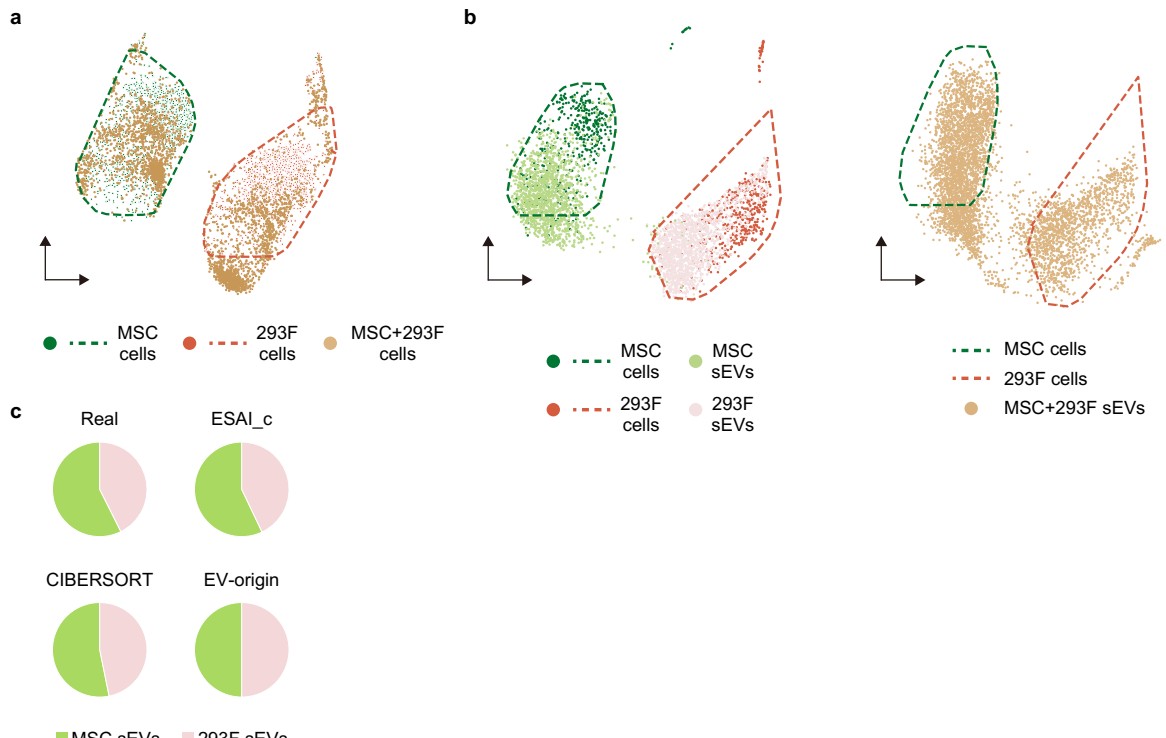

**Extended Data Fig. 6 | SEVtras can reveal the heterogeneity of sEVs with broad applicability. a**, Cell states in separated and mixed MSC and 293F samples. Cells of mixture sample were subsampled to 3,000. **b**, sEV-containing droplets in mixture sample were highly concordant with sEV-specific clusters of MSC and 293F after BBKNN batch adjustment (see Methods). Droplets were subsampled to 6,000 for visualization. **c**, Benchmark the performance of SEVtras and the state-of-the-art methods (CIBERSORT and EV-origin) in sEV composition deconvolution. scRNA-seq data of MSC and 293F mixture was used as the benchmark dataset.

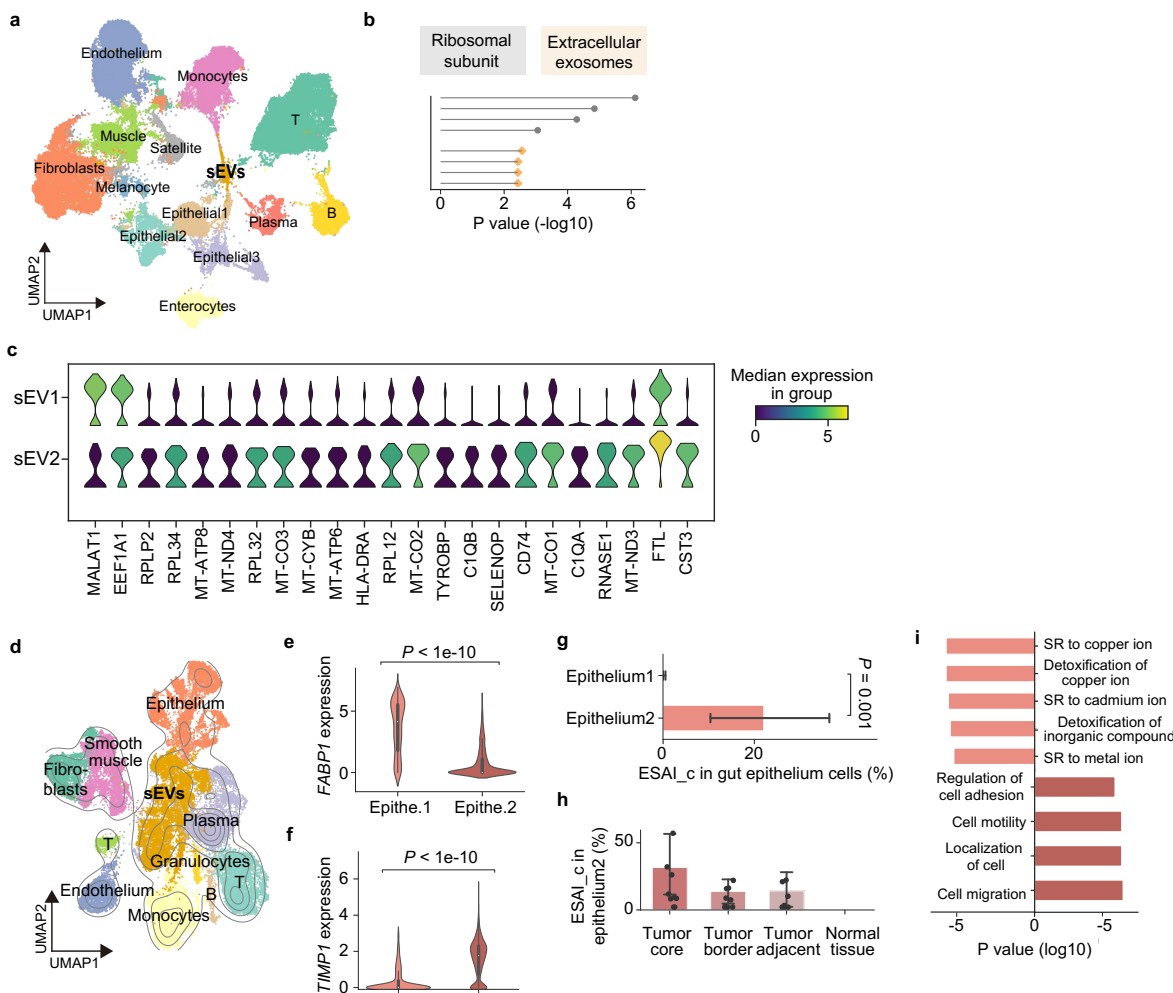

**Extended Data Fig. 7 | SEVtras deepens single cell transcriptome analysis.**
**a**, UMAP of identified sEV-containing droplets and cell types in 15 human normal tissues dataset. **b**, GO enrichment of droplets identified by SEVtras in 15 human normal tissues dataset. **c**, Genes enriched in the two sEV subtypes in 15 human normal tissues dataset. **d**, UMAP of identified sEV-containing droplets and cell types in CRC dataset. **e**, Expression of *FABP1* in the two gut epithelial subtypes in CRC dataset ($n = 5,066$). '*P*' represents the *P* value in two-sided Mann-Whitney U test. Data are shown as median values with interquartile range. **f**, Expression of *TIMP1* in the two gut epithelial subtypes in CRC dataset ($n = 5,066$). '*P*' represents the *P* value in two-sided Mann-Whitney U test. Data are shown as median values with interquartile range. **g**, ESAI_c in the two gut epithelial subtypes in CRC dataset ($n = 36$). '*P*' represents the *P* value in two-sided Mann-Whitney U test. The bar represents the mean, and the lower and upper limits in the error bar are the values corresponding to the 2.5th and 97.5th percentiles after 1,000 bootstrap iterations. **h**, ESAI_c of gut epithelium2 at different locations relative to the tumor in CRC dataset ($n = 27$). The bar represents the mean, and the lower and upper limits in the error bar are the values corresponding to the 2.5th and 97.5th percentiles after 1,000 bootstrap iterations. **i**, GO enrichment of differentially expressed genes in the two gut epithelial subtypes in CRC dataset. '*P*' represents the *P* value in GO enrichment using a hypergeometric test.

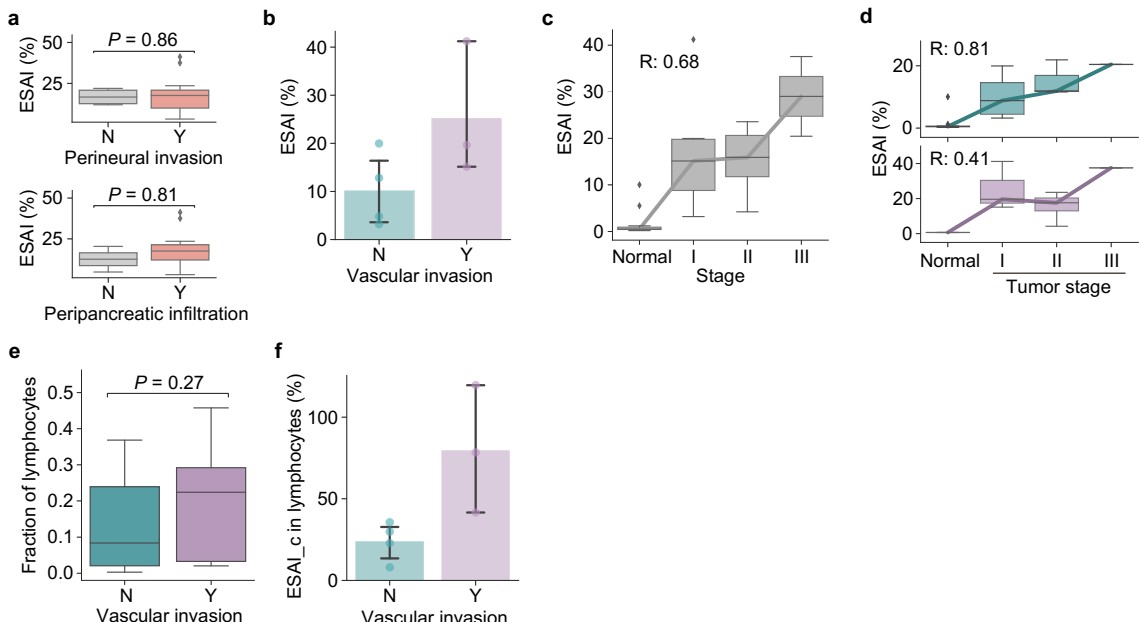

**Extended Data Fig. 8 | ESAI_c is strongly associated with vascular invasion in early-stage tumors. a**, The relationship between ESAI and perineural invasion or peripancreatic infiltration (*n* = 27). '*P*' represents the *P* value in two-sided Mann-Whitney U test. Data are shown as median values with interquartile range. **b**, ESAI correlates with the tumor vascular invasion in the stage I (*n* = 18). The bar represents the mean, and the lower and upper limits in the error bar are the values corresponding to the 2.5th and 97.5th percentiles after 1,000 bootstrap iterations. **c**, ESAI across different tumor stage (*n* = 27). Data are shown as median values with interquartile range. **d**, ESAI across different tumor stages, separated by vascular invasion status (*n* = 27). Data are shown as median values with interquartile range. **e**, The relationship between the fraction of lymphocytes and the tumor vascular invasion (*n* = 27). '*P*' represents the *P* value in two-sided Mann-Whitney U test. Data are shown as median values with interquartile range. **f**, ESAI_c of lymphocytes correlates with the tumor vascular invasion in the stage I (*n* = 18). The bar represents the mean, and the lower and upper limits in the error bar are the values corresponding to the 2.5th and 97.5th percentiles after 1,000 bootstrap iterations.

# Reporting Summary

## Statistics

For all statistical analyses, confirm that the following items are present in the figure legend, table legend, main text, or Methods section.

| n/a | Confirmed | |
|---|---|---|
| ☐ | ☒ | The exact sample size (*n*) for each experimental group/condition, given as a discrete number and unit of measurement |
| ☐ | ☒ | A statement on whether measurements were taken from distinct samples or whether the same sample was measured repeatedly |
| ☐ | ☒ | The statistical test(s) used AND whether they are one- or two-sided<br>*Only common tests should be described solely by name; describe more complex techniques in the Methods section.* |
| ☒ | ☐ | A description of all covariates tested |
| ☐ | ☒ | A description of any assumptions or corrections, such as tests of normality and adjustment for multiple comparisons |
| ☐ | ☒ | A full description of the statistical parameters including central tendency (e.g. means) or other basic estimates (e.g. regression coefficient) AND variation (e.g. standard deviation) or associated estimates of uncertainty (e.g. confidence intervals) |
| ☒ | ☐ | For null hypothesis testing, the test statistic (e.g. *F*, *t*, *r*) with confidence intervals, effect sizes, degrees of freedom and *P* value noted<br>*Give P values as exact values whenever suitable.* |
| ☒ | ☐ | For Bayesian analysis, information on the choice of priors and Markov chain Monte Carlo settings |
| ☒ | ☐ | For hierarchical and complex designs, identification of the appropriate level for tests and full reporting of outcomes |
| ☐ | ☒ | Estimates of effect sizes (e.g. Cohen's *d*, Pearson's *r*), indicating how they were calculated |

*Our web collection on statistics for biologists contains articles on many of the points above.*

## Software and code

Policy information about availability of computer code

| Data collection | Public data were downloaded from NCBI/ArrayExpress/NGDC/customed websites through wget (v1.12). |
|---|---|
| Data analysis | Our proposed method SEVtras is available at https://github.com/bioinfo-biols/SEVtras (v 0.3). For RNA-seq data, raw reads were cleaned using Trim Galore (v0.6.7) and aligned to the GRCh38 human reference genome using STAR (v2.6.1a) and RSEM (v1.2.25) to quantify the transcripts. For scRNA-seq data, raw reads were processed by Cell Ranger (v5.0.0) and downstream single-cell analysis was performed with Scanpy (v1.8.2), MAGIC (v3.0.0) and BBKNN (v1.5.1). All data were analyzed using python 3.8 with numpy (v1.20.3) and pandas (v1.2.4), and visualized using matplotlib (v3.4.2) and seaborn (v0.11.0). |

For manuscripts utilizing custom algorithms or software that are central to the research but not yet described in published literature, software must be made available to editors and reviewers. We strongly encourage code deposition in a community repository (e.g. GitHub). See the Nature Portfolio guidelines for submitting code & software for further information.

## Data

Policy information about availability of data

All manuscripts must include a data availability statement. This statement should provide the following information, where applicable:

- Accession codes, unique identifiers, or web links for publicly available datasets
- A description of any restrictions on data availability
- For clinical datasets or third party data, please ensure that the statement adheres to our policy

The bulk RNA-seq and scRNA-seq data for MSC and 293F cells were deposited at NGDC with accession number PRJCA017291 (https://ngdc.cncb.ac.cn/gsa-human/ browse/HRA004708). The CITE-seq data and scRNA-seq data for normal tissues and prostate cancer were downloaded from the NCBI Gene Expression Omnibus (GSE150599, GSE159929 and GSE137829, respectively). The scRNA-seq data for colorectal cancer were accessed in ArrayExpress under accession number E-MTAB-8410. The scRNA-seq data for pancreatic ductal adenocarcinoma were accessed in GSA with the accession number CRA001160. The scRNA-seq data for gastric cancer were accessed at https://dna-discovery.stanford.edu/research/datasets/.

## Human research participants

Policy information about studies involving human research participants and Sex and Gender in Research.

| | |
|---|---|
| Reporting on sex and gender | The study did not involve sex and gender. |
| Population characteristics | The study did not recruit participants, as we used public datasets from previous studies. |
| Recruitment | The study did not recruit participants, as we used public datasets from previous studies. |
| Ethics oversight | The study did not involve ethics, as we used public datasets from previous studies. |

Note that full information on the approval of the study protocol must also be provided in the manuscript.

# Field-specific reporting

Please select the one below that is the best fit for your research. If you are not sure, read the appropriate sections before making your selection.

☒ Life sciences ☐ Behavioural & social sciences ☐ Ecological, evolutionary & environmental sciences

For a reference copy of the document with all sections, see nature.com/documents/nr-reporting-summary-flat.pdf

# Life sciences study design

All studies must disclose on these points even when the disclosure is negative.

| | |
|---|---|
| Sample size | No sample size calculations were performed. For RNA-seq, we only used cell lines for sequencing and all samples were replicated with two biological replicates to ensure reproducibility. For scRNA-seq, we first only performed on cell lines, all of which had >1,000 cells. We then collected six public large-scale scRNA-seq datasets comprising > 90 samples, which should be sufficient as in most single-cell studies. |
| Data exclusions | No data were excluded from the analyses. |
| Replication | We generated two biological replicates for all conditions explored in our study. All attempts were confirmed to be successful. |
| Randomization | Randomization was not relevant to our study, as we used public datasets from previous studies and no group assignment was needed. |
| Blinding | Blinding was not necessary for the same reason as no objective scoring was applied in our study. |

# Reporting for specific materials, systems and methods

We require information from authors about some types of materials, experimental systems and methods used in many studies. Here, indicate whether each material, system or method listed is relevant to your study. If you are not sure if a list item applies to your research, read the appropriate section before selecting a response.

## Materials & experimental systems

| n/a | Involved in the study |
|---|---|
| ☐ | ☒ Antibodies |
| ☐ | ☒ Eukaryotic cell lines |
| ☒ | ☐ Palaeontology and archaeology |
| ☒ | ☐ Animals and other organisms |
| ☒ | ☐ Clinical data |
| ☒ | ☐ Dual use research of concern |

## Methods

| n/a | Involved in the study |
|---|---|
| ☒ | ☐ ChIP-seq |
| ☒ | ☐ Flow cytometry |
| ☒ | ☐ MRI-based neuroimaging |

# Antibodies

| | |
|---|---|
| Antibodies used | The antibodies used for western blot:<br>1. Rabbit polyclonal anti-CD9, Proteintech, Cat# 20597-1-AP.<br>2. Rabbit polyclonal anti-Syntenin-1, Proteintech, Cat# 22399-1-AP.<br>3. Rabbit polyclonal anti-Calnexin, EASYBIO, Cat# BE3386.<br>4. Rabbit polyclonal anti-GRP94, Proteintech, Cat# 14700-1-AP.<br>5. HRP-conjugated goat anti-rabbit antibody, EASYBIO, Cat# BE0101. |
| Validation | Validation was relied on the available data provided by the manufacture's websites for all antibodies:<br>1. Proteintech claims that this Rabbit polyclonal anti-CD9 (Cat# 20597-1-AP) is suitable for WB applications in human. (https://www.ptgcn.com/products/CD9-Antibody-20597-1-AP.htm)<br>2. Proteintech claims that this Rabbit polyclonal anti-Syntenin-1 (Cat# 22399-1-AP) is suitable for WB applications in human. (https://www.ptgcn.com/products/SDCBP-Antibody-22399-1-AP.htm)<br>3. EASYBIO claims that this Rabbit polyclonal anti-Calnexin (Cat# BE3386) is suitable for WB applications in human. (http://bioeasytech.com/product/2629.html?goods_id=4824)<br>4. Proteintech claims that this Rabbit polyclonal anti-GRP94 (Cat# 14700-1-AP) is suitable for WB applications in human. (https://www.ptgcn.com/products/HSP90B1-Antibody-14700-1-AP.htm)<br>5. EASYBIO claims that this HRP-conjugated goat anti-rabbit antibody (Cat# BE0101) is suitable for WB applications to detect rabbit primary antibody. (http://www.bioeasytech.com/product/2901.html?goods_id=5786) |

# Eukaryotic cell lines

Policy information about cell lines and Sex and Gender in Research

| | |
|---|---|
| Cell line source(s) | MSC cells were provided by Jinan Wanquan Biotechnology (Shandong, China). 293F cells were originally acquired from QuaCell Biotechnology (Guangdong, China). |
| Authentication | MSC and 293F cell lines were authenticated by STR DNA profiling analysis. |
| Mycoplasma contamination | MSC and 293F cell lines were tested negative for mycoplasma contamination. |
| Commonly misidentified lines<br>(See ICLAC register) | No commonly misidentified cell line was used in this study. |

