## [Peer Review File · Nature Methods]

Peer Review Information

Manuscript Title: SEVtras delineates small extracellular vesicles at droplet resolution from single-cell transcriptomes

Corresponding author name(s): Peifeng Ji, Fangqing Zhao

Editorial Notes: n/a

Reviewer Comments & Decisions:

Decision Letter, initial version:

Dear Professor Zhao,

Your Brief Communication entitled "exoTras delineates exosome profiles at droplet resolution from single-cell transcriptomes" has now been seen by 4 reviewers, whose comments are attached. While they find your work of some potential interest, they have raised serious technical concerns which in our view are sufficiently important that they preclude publication of the work in Nature Methods.

We will consider looking at a revised manuscript only if further experimental data allow you to address all the major criticisms of the reviewers (unless, of course, something similar has by then been accepted at Nature Methods or appeared elsewhere). This includes submission or publication of a portion of this work somewhere else.

The required new experiments and data include, but are not limited to evidence that exoTras is analyzing true EVs. We hope you understand that until we have read the revised paper in its entirety we cannot promise that it will be sent back for peer-review.

If you are interested in revising this manuscript for submission to Nature Methods in the future, please contact me to discuss your appeal before making any revisions. Otherwise, we hope that you find the reviewers' comments helpful when preparing your paper for submission elsewhere.

Sincerely,
Madhura

Madhura Mukhopadhyay, PhD
Senior Editor
Nature Methods

Reviewers' Comments:

Reviewer #1:

Remarks to the Author:

He et al. describes a new algorithm (exoTras) to identify exosome-containing droplets in droplet-based single cell RNA sequencing (scRNA-seq). The algorithm was designed to count exosome-containing droplets, estimate exosome secretion activity and trace their secretory cell types. The authors then applied the algorithm to tumor scRNA-seq datasets to evaluate potential clinical applications. Overall the study is interesting. The authors need to clarify the advantages of their approach and perform extensive control experiments for the current claims.

1) The introduction does not reflect recent advances in single cell-based extracellular vesicle profiling and tracking of vesicles to cells of origin. The authors should compare the current approach with relevant published studies.

<https://www.science.org/doi/10.1126/sciadv.abb3461>

<https://www.pnas.org/doi/10.1073/pnas.2108876118>

<https://www.pnas.org/doi/abs/10.1073/pnas.1814348116>

<https://www.sciencedirect.com/science/article/pii/S2001037020304244>

2) The authors need to clarify the advantages of their approach (profiling exosomes at the droplet resolution). While several published studies have investigated single-cell extracellular vesicle secretion (see above), the current study is not really set up to measure specific vesicle secretion. Since it uses published scRNA-seq data, depending on the experimental preparation for these datasets, multiple unrelated vesicles could be encapsulated within a single droplet. Given the large size of the droplets with respect to exosome size, the study is essentially performing relatively bulk exosome profiling through droplet sampling. How is the approach advantageous as compared to dedicated single-cell vesicle secretion study as well as bulk exosome RNA profiling?

3) The development of the EM framework and unified gene set is interesting but requires more rigorous validation. Currently, the experimental exosome-exclusive data was generated from total RNA seq (not from droplet-based sequencing). Additional studies are needed to answer key questions such as the

minimal exosome counts per droplet required for accurate counting and classification, algorithm performance against complex background (other cell-free RNAs also encapsulated in the same droplets).

4) For exosome source tracking, the authors made several assumptions (1) the transcriptional profile of an exosome is more similar to its cell of origin and (2) exosome release is affected by the biogenesis capacity of the original cell type. How generalizable are these assumptions across datasets?

5) Since multiple vesicles are likely to be encapsulated within a single droplet, for droplets that contain heterogeneous mixture of exosomes derived from different cellular origins, how were they classified?^[1]_{SEP}

6) How accurate is the approach as compared to other published tracking methods? Further investigation should be performed to assess the tracking accuracy, using pure vesicle populations derived from single cell source as well as known mixtures prepared with vesicles derived from different cells of origin.

7) Induction and inhibition of exosome secretion could be included in multiple cell types to evaluate the responsiveness of ESAI.

Reviewer #2:

Remarks to the Author:

The authors present exoTras, a workflow that identifies exosomes containing droplets from single cell RNA-seq datasets. The algorithm is unique, novel and much needed to the field of extracellular vesicles (EVs). Though this reviewer is supportive of this article to be published, certain parts of the manuscript need to be updated. Each section needs more clarity as what was done and what is reported in the figures. Now it is quite vague and hard to understand.

1. The authors need to add additional data on the characterization of exosomes/EVs from single cell seq datasets and their workflow. What controls are in place to ensure that these are indeed EVs and not cell debris? Perhaps the authors can compare cells, debris (intentionally obtained with harsh treatment or stress), EVs for EV markers (CD63, CD81, Alix, Sytenin, TSG101) and negative (GRP96, Calnexin). This can be done on A549 cell dataset (Figure 1c) as well. The cells can be broken into debris by passing them through sieve or mechanical stress and compared with EVs and cell lysates.

2. What quality control is placed to check that the sample used does not suffer from damage? Is there a possibility of including a negative control? Perhaps a RNA profile signature from debris?

3. ESAI is very interesting and informative. Though this is very useful, what if the EVs are migrating from adjacent cells? How does the algorithm tackle this? Hence the ESAI should also consider the cell type source for EVs and their locality in the tissue source of interest.

4. Throughout the manuscript it is not clear what data is generated by the authors and those that are utilised from publicly available resources. Perhaps the authors can clarify in the manuscript. This also brings a point that the authors need to do this comprehensively on one dataset that they generated inhouse – where all characterization such as western blot for markers, quality control, debris all needs to be done. The existence of two exosomes group can also be shown comprehensively. The authors can apply exoTras to other datasets later like in the manuscript and shows its utility.
5. What does ESAI of 8.3% mean? Does it mean X of exosomes droplets / Y of total cells? How does the authors deal with EV subtypes and heterogeneity here? What about non-exosomes droplets but not cell droplets? The manuscript lacks clarity at various levels for every section.
6. ESAI in figure 2e. How was this done? The legend is not clear? What does the error bar mean? Why is the error bar for Tumor adjacent not equal on both sides?
7. The use of the term “exosome” need to be avoided throughout the manuscript including figures. Exosome refer to the RNA complex while exosomes (plural) refers to endocytic driven extracellular vesicles.
8. The authors should use the term extracellular vesicles (EVs) or small EVs in general as opposed to exosomes. Exosomes are endocytically derived sEVs but no current technology can purify exosomes alone to 100% purity.

Reviewer #3:

Remarks to the Author:

The manuscript by He et al proposes a new algorithm ('exoTras') via which the RNA content and potential tissue origin of 'exosomes' (a subtype of extracellular vesicles) may be assessed. Hereto, the authors re-analyze droplet-based single cell RNAseq datasets. Droplet sequencing of (single) extracellular vesicles (EVs) isolated from culture medium or bodyfluids has been performed previously. However, these previous approaches did not allow tracing back the origin of the EVs to their secretory cell types.

The new approach proposed here is based on the idea that libraries of droplet-based scRNAseq experiments are composed of cell-containing droplets, vacant droplets, cellular debris-containing droplets and exosome-containing droplets. It took me some time to understand the conceptual background of this approach, but I presume that it is based on the idea that droplets that contain RNA but no cells are thought to contain extracellular RNA (e.g. enclosed in EVs) released by the cells in question. I have doubts on the foundation of this approach based on the following arguments:

1. It is highly unlikely that the material that is found in non-cell-containing droplets of the scRNAseq experiments are actively released extracellular vesicles. The cells (either from cell cultures or detached from tissues by enzymatic treatments) are processed by washing and pelleting, and in some cases were subsequently cryopreserved. There was no recovery / culture step just prior to entering cells in the

scRNAseq system (e.g. Chromium system). Therefore there has been no time/good condition for cells to actively release natural EVs. It is therefore far more likely that the extracellular RNAs are from rip-offs, which indeed would reflect the RNAome of the cells that had been processed.

In relation to the previous point, it is very difficult to substantiate that the observed differences in the amount of 'exosome' material between different (parts of) tissues is really indicative of differences in the amount of EVs these tissue cells release (indicated by the authors as the ESAI: 'exosome secretion activity index'). The differences may just be a consequence of differences in the fragility of different cell types or the difficulty with which each cell type can be freed from the tissue environment.

2. The authors claim to be able to differentiate between 'debris' and 'exosomes'. First, it is currently not possible to differentiate between endosome-derived EVs (exosomes) and other types of actively released EVs once they are present outside cells. There is no evidence in this manuscript that the observed material is from genuine exosomes. Second, the adopted approach to differentiate between 'debris' and 'exosomes' is not sound. The reference EV transcriptome has been composed by manually curating EV-derived RNA gene sets from online available databases (exoCarta, ExRNA atlas). Especially the exoCarta database contains data of studies where only an enrichment step has been performed to isolate EVs. These samples are highly contaminated with non-EV-associated RNAs and therefore not suitable to use as specific reference for EV (or exosome)-associated RNAs. This, in combination with the previous point, severely complicates definitions of 'exosomes' versus 'debris'.

3. It is unclear how is assessed whether there is one or multiple EVs in a drop and the interpretation thereof (e.g. are these multiple EVs from one or multiple cell types?).

4. With regard to the previous arguments, the manuscript would have largely benefited from including solid experimental validation. Now, most of the 'validations' are simulations, which are not sufficient for substantiating the credibility of this entirely new approach. Experimental validation for assessing whether actively released EVs are measured and for the ESAI could for example be a droplet scRNAseq analysis of cultured cells treated or not with bafilomycin A1 as a reagent known to induce increased EV release. In addition, the capacity of exoTras to distinguish EV- versus debris-associated RNA could be validated by performing droplet scRNAseq of cells spiked with variable ratios of highly purified EVs and mechanically desintegrated cells (i.e. 'debris').

Additional comment:

The authors should clarify in more detail what the application area of the ESAI and 'EV' typing is: do they expect that this exoTras method can be used for diagnosis of cancer, and which input material would be

required for this? Is it expected that exoTras data reveal disease-associated exRNA transcriptomes that might be traced back in body fluids used as liquid biopsies?

Reviewer #4:

Remarks to the Author:

Exosomes are extracellular vesicles involved in the intercellular transportation of materials including nucleic acids, proteins, and/or other regulatory factors. They play vital roles in cell-cell communications regulating development, immune response, and disease onset and progression, and have been engineered as a vehicle for drug delivery. To elucidate the heterogeneity of exosome cargos, to identify their source of origin, and to unravel the regulatory roles of different types of exosomes have attracted much research.

The study by He et. al very nicely mounted to the ample availability of droplet-based single cell sequence data to develop an analytic tool set, exoTras, for exosome analysis. The main contribution includes a few algorithms to identify exosome containing droplets, to estimate exosome secretion activity for different types of cells, and to track back to their secretory cell types. The study also applied exoTras to analyze the droplet scRNA-seq data for normal tissues and cancer samples, and found exosome subtypes of different cargo type and cell source, as well as the elevated exosome secretion activity of aggressive epithelial cells in CRC and the potential usage of exosomes in biomarker for vascular invasion in early-stage cancer development.

This exoTras tool could be a powerful one that can dig into huge amount existing data from of droplet-based scRNA-seq studies (and requires no additional invest in new studies). The following are my comments:

1, it is not clear from the description how the authors deal with droplets of mixed content of cell debris and exosomes. I guess the authors know this, but the writing gives no clue on how these are handled and how it will affect the performance of exoTras.

2, the normalization of different samples is not well-described in the manuscript. How is the sequencing depth handled as a factor? I have a concern looking at Figure 1b--apparently total UMI affects the AUC. What about the other types of confounding factors considered? Some sample preparation protocols may deplete exosomes and the others may enrich them. Some experimental settings may favor a droplet to contain more exosomes and the others may disfavor it. The sequencing platform may favor the obtaining of specific exosome content. All of these factors may affect the sequencing results; it is very much like batch effects in single-cell data analysis. Without careful normalization and batch effect removal, it is dangerous to compare across samples.

This is especially a factor that needs to be carefully taken care of for the cancer applications in page 7.

3, in the simulation study, the cell-exclusive sample are likely to be associated with exosomes as well. Please be precise. Also, will this be a problem? See above, what if the fraction of associated exosomes varies? Will this be a confounding factor?

4, figure 1c, the correlation coefficients seem low even for droplets. This is rather expected given the little amount of material in each droplet and the data loss in scRNA-seq. Would exoTras benefit from data imputation?

5, the study only investigated A549 cells. It would be very nice if the authors can perform more cell lines and to compare. It will also help the cell source tracing (for example, being a validation).

6, citation to supplementary figures should also be clear onto panels.

Minor

1, page 3, middle paragraph, I do not agree that the decrease of specificity is slight.

2, page 5~6, the data are from scRNA-seq studies of different tissues/organs, and the analyses are on constituent cell types. But the writing is a little confusing without the introduction of cell types. I made a mistake at the first time thinking that the authors confuse tissues and cell types. Better to mention the two to be less ambiguous.

Author Rebuttal to Initial comments

Dear Madhura,

Thank you very much for considering our paper and providing us an opportunity to address the concerns and comments raised by the four reviewers, whose specific comments and suggestions provided valuable feedback helping us improve our study.

We have addressed all of the reviewers' concerns with a substantial amount of new data and analyses.

Our responses are composed of two parts: **overall responses** and **point-by-point responses**. We first address the common concerns raised by all reviewers in the **overall responses**. Then we reply the specific comments in the **point-by-point responses**. Based on the suggestion of *Reviewer #2*, we have changed the name of our method from "exoTras" to "SEVtras".

Given the growing significance and abundance of scRNA-seq datasets, we are confident that SEVtras, which provides extracellular insights and possesses the potential to unlock valuable diagnostic applications, will be highly appealing to the readers of Nature Methods.

Thank you very much for your consideration.

Best regards,

Fangqing Zhao

Beijing Institutes of Life Science, Chinese Academy of Sciences

Overall responses:

We have extensively revised our manuscript based on the four major concerns raised by the reviewers.

Q1) How to ensure that SEVtras indeed characterizes sEVs?

All the reviewers have raised the same concern that whether our algorithm indeed characterized sEVs (*Reviewer #1 comment 3, Reviewer #2 comments 1-2, Reviewer #3 comments 2 and 4, Reviewer #4 comment 1*). In response, we have designed new bulk RNA-seq and scRNA-seq experiments using two cell lines (MSC and 293F) to address this issue.

We first generated sEV-exclusive bulk RNA-seq datasets by utilizing experimentally isolated sEVs from each cell line. Subsequently, we calculated the gene expression correlation between the sEV-exclusive bulk data and the sEV-containing droplets identified by SEVtras, as well as the debris-containing droplets identified by SEVtras. As expected, we found a significantly increased correlation between the former two datasets compared to the correlation between the sEV- and debris-containing droplets identified by SEVtras ($P < 0.05$, T-test) (**Supplementary Figure 3ab**).

To further assess to what extent the sEVs identified by SEVtras are similar with the real ones, we performed scRNA-seq on experimentally isolated sEVs obtained from the two cell lines as ground truth. We found a high expression correlation between the ground truth dataset and sEV-containing droplets identified by SEVtras (Pearson correlation $R = 0.77$ and 0.86 for MSC and 293F, respectively) (**Response Figure 1**). However, the correlation between the ground truth dataset and debris-containing droplets was notably low (Pearson correlation $R = 0.16$ and 0.11 for MSC and 293F, respectively). To further reinforce this observation, we performed UMAP analysis, which demonstrated that the distribution of sEV-containing droplets overlapped with that of experimentally isolated sEVs, while differing from the distribution of debris (**Response Figure 2**). Taken together, our results demonstrate that SEVtras can reliably detect sEV signals from scRNA-seq datasets.

Response Figure 1. sEVs identified by SEVtras were significantly correlated with experimentally isolated sEVs in both MSC and 293F cell lines. Here, we added up RNA counts of experimentally isolated sEVs in the ground truth data as pseudo-bulk sample for comparison. Left panel: Pearson correlation between droplets identified by SEVtras and the pseudo-bulk data in MSC and 293F cell lines. Droplets were classified into two classes: sEV-containing (yellow) and debris-containing (grey) in the boxplot. Right panel: matrixes of debris and sEVs were imputed by MAGIC (Van Dijk et al., 2018, Cell).

Response Figure 2. sEVs identified by SEVtras were highly overlapped with experimentally isolated sEVs in the UMAP analysis in both MSC and 293F cell lines. Left panel: workflow of ground truth data generation. Right panel: sEV-containing droplets identified by SEVtras matches the location of ground truth data in UMAP. Blue and grey scatters represent sEV- and debris-containing droplets identified by SEVtras, respectively. Grey and yellow and lines represent the contour map of the density of debris and experimentally isolated sEVs, respectively.

We have included these new results as **Extended Data Fig. 3b** and **Fig. 1b** in the revised manuscript.

Q2) Whether SEVtras is influenced by cell debris and large EVs?

The reviewers also expressed concerns on the accuracy of SEVtras, which might be affected by cell debris and large EVs (*Reviewer #1 comment 3, Reviewer #2 comments 1-2, Reviewer #3 comments 2 and 4, Reviewer #4 comment 1*). We address this issue by 1) comparing the gene expression of sEVs with that of cell debris and large EVs, 2) spiking cell debris and large EVs into scRNA-seq samples.

First, to determine whether specific genes exhibit enrichment in sEVs, we generated distinct bulk RNA-seq datasets, namely cell-exclusive, debris-exclusive and large EV-exclusive datasets, using MSC and 293F cell lines, respectively (**Response Figure 3a**). Subsequently, we compared the gene expression profiles of the sEV-exclusive bulk RNA-seq data with those of the cell debris and large EVs (**Response Figure 3bc**). We found that previously collected sEV-gene set was significantly enriched in the sEV-exclusive dataset compared with the exclusive datasets of cell

debris and large EVs across both cell lines ($P < 0.05$, GSEA). This high enrichment serves as a foundation for distinguishing signals of sEVs from unrelated cell compartments.

Response Figure 3. Transcriptional profile of sEVs is distinct with that of cell debris and large EVs in both MSC and 293F cell lines. **a**, Experimental design of generating exclusive transcriptional profiles of debris, sEVs and large EVs (IEVs). Debris, sEVs and IEVs are isolated from cell cultures. In particular, for cell debris, we mechanically ground MSC/293F cells with four minutes (Supplementary Figure 1a-c). For IEVs, we centrifuged MSC/293F cell medium at 2,000 \times g and collected pellets (see Methods). sEVs is separated by ultracentrifugation (see Methods). **b-c**, Comparisons between sEVs and debris or debris (IEVs) in MSC (**b**) and 293F (**c**) cell lines. Left (ternary plot): the expression of each gene in cell, sEVs and debris or IEVs. Color depicts the fold change between sEVs and debris or IEVs. Right (GSEA plot): enrichment score of the sEVs-gene set in sEVs compared to debris or IEVs. “P” means FWER P values in the GSEA.

Second, to further test the robustness of SEVtras in the presence of complex backgrounds, we conducted spike-in experiments involving cell debris and large EVs generated from approximately 10,000 MSC cells. These two types of spikes were separately added to the MSC single-cell suspension, followed by scRNA-seq analysis and SEVtras identification (Response Figure 4). Interestingly, we observed no significant change in the ESAI of either spike-in sample compared to the ESAI of the untreated MSC sample ($P > 0.05$, T-test). Moreover, the gene expression profiles of sEVs in both spike-in samples exhibited a high correlation with the untreated sample (Extended Data Fig. 3cd) (MSC + debris: Pearson correlation $R = 0.999$) (MSC +

large EVs: Pearson correlation $R = 0.997$). Thus, SEVtras can specifically identify the signals of sEVs.

Response Figure 4. ESAI could specifically identify the signals of sEVs. Left panel, schema of the spike-in experiments in MSC cell line. In particular, Cell debris and large EVs (IEVs) were generated from approximately $1e4$ MSC cells, respectively. For cell debris, we mechanically ground MSC cells with four minutes (**Supplementary Figure 1a-c**). For IEVs, we centrifuged MSC cell medium at $2,000 \times g$ and collected pellets (see Methods). Right panel, ESAI of these spike-in samples with different treatments.

We have included these new results as **Extended Data Fig. 2a-c and 3c** and **Fig. 1c** in the revised manuscript.

Q3) How to evaluate the reliability of ESAI?

In addition to the identification accuracy, the reviewers also want to know the reliability of ESAI (*Reviewer #1 comment 7, Reviewer #3 comments 1 and 4*). In the revised manuscript, we have designed new scRNA-seq experiments to disturb the amount of sEVs.

To qualitatively assess the reliability of ESAI, we stimulated the secretion activity of MSC by introducing monensin sodium salt (MON) to the cultured MSC cells (**Response Figure 5**). Additionally, we incorporated isolated sEVs into the single-cell suspension to further evaluate the assay (**Response Figure 5**). After implementing SEVtras, we observed an enhanced ESAI in the two samples compared with untreated MSC cells (**Response Figure 5**). To further stimulate the “secretion” of sEVs, we treated MSC cell line with MON and simultaneously introduced one-third of the isolated sEVs. As a result, the ESAI of this sample increased by 234% compared to the sample treated with MON alone, approaching the one-third increase observed when only isolated sEVs were added (241%). Based on these findings, we confidently conclude that ESAI accurately reflects the sEV secretion activity of cells.

Response Figure 5. ESAI can faithfully reflect sEV secretion activity of MSC cells. Left panel, schema of the spike-in experiments in MSC cell line. In particular, Cell debris and large EVs (IEVs) were generated from approximately $1e4$ MSC cells, respectively. For cell debris, we mechanically ground MSC cells with four minutes (Supplementary Figure 1a-c). For IEVs, we centrifuged MSC cell medium at $2,000 \times g$ and collected pellets (see Methods). sEVs was spiked in the scRNA-seq sample with a total of $3e7$ particles. Monensin sodium salt was added at a dose of $5 \mu\text{M}$ for 18 hours. Right panel, ESAI of these spike-in samples with different treatments.

We have added this data as Fig. 1c in the revised manuscript.

Q4) How to assess the sEV deconvolution accuracy?

Regarding the accuracy of sEV deconvolution for different cell types (Reviewer #1 comments 4-6, Reviewer #3 comment 1, Reviewer #4 comment 5), we first compared the gene expression profiles of sEVs between MSC and 293F, and then designed additional scRNA-seq experiments by mixing the two cell lines.

To reveal the heterogeneity present in the sEVs originating from different cell types, we conducted UMAP analysis to compare their transcriptional profiles between MSC and 293F cell lines (Response Figure 6a). Our analysis revealed a distinct separation between the sEVs derived from these two cell lines. However, it should be noted that the sEVs from both cell lines overlapped with their respective original cells. This finding indicates that sEVs originating from diverse cell types possess distinct features, thereby forming the basis for the accurate deconvolution of sEV secretion activity associated with different cell types within the complex microenvironment of tissues.

To further validate the accuracy of ESAI in a complex scenario, we conducted scRNA-seq on a mixed sample containing cells from both MSC and 293F cell lines at a 1:1 ratio (Response Figure 6b). Remarkably, the ESAI of this mixed sample (320%) was consistent with the averaged value of ESAI for MSC and 293F (322%). Importantly, UMAP analysis showed that the sEV-containing droplets from this mixture formed two

distinct and heterogeneous clusters. These two clusters of sEVs were highly similar to the clusters observed in the analysis of sEVs and original cells described above (**Response Figure 6a**).

Based on these findings, we were able to employ the identified sEV-containing droplets to deconvolve the sEV secretion activity to certain cell type, and calculate their ESAI (herein ESAI_c) (**Response Figure 6c**). We found that ESAI values for MSC and 293F were 359% and 276%, respectively. These values were close to the ESAI values obtained for each cell line alone (MSC: 369%, 293F: 274%). Collectively, these results demonstrate the ability of SEVtras to accurately decipher the sEV secretion activity of different cell types.

Response Figure 6. SEVtras can accurately decipher sEV secretion activity of different cell types. **a**, Left panel, workflow for scRNA-seq data generation of MSC/293F cell line. Right panel, the heterogeneity of sEVs in different cell types. **b**, Left panel, workflow for scRNA-seq data generation of MSC and 293F mixtures. Right panel, sEV-containing droplets were highly concordant with previous sEV-specific clusters of MSC and 293F. **c**, Top panel, schema of deconvolving ESAI for different cell types. ESAI_c, ESAI in certain cell type. Bottom panel, ESAI can accurately decipher the sEV secretion activity of different cell types.

We have added these results as **Fig. 2a-c** in the revised manuscript.

Point-by-point responses:

Below is a point-by-point detailed response to the reviewers' comments. The comments are reproduced, and our responses are given directly afterward in a different color (black).

Reviewer #1

Remarks to the Author:

He et al. describes a new algorithm (exoTras) to identify exosome-containing droplets in droplet-based single cell RNA sequencing (scRNA-seq). The algorithm was designed to count exosome-containing droplets, estimate exosome secretion activity and trace their secretory cell types. The authors then applied the algorithm to tumor scRNA-seq datasets to evaluate potential clinical applications. Overall the study is interesting. The authors need to clarify the advantages of their approach and perform extensive control experiments for the current claims.

Response: We greatly appreciate the reviewer's comments on the novelty and significance of our study. We have clarified the advantages of our approach in the revised manuscript and conducted additional control experiments.

1) The introduction does not reflect recent advances in single cell-based extracellular vesicle profiling and tracking of vesicles to cells of origin. The authors should compare the current approach with relevant published studies.

<https://www.science.org/doi/10.1126/sciadv.abb3461>

<https://www.pnas.org/doi/10.1073/pnas.2108876118>

<https://www.pnas.org/doi/abs/10.1073/pnas.1814348116>

<https://www.sciencedirect.com/science/article/pii/S2001037020304244>

Response: Thanks for the insightful comments. We have added additional text in the first paragraph (Introduction) of the revised manuscript and copied as follow:

"Recently, several studies have attempted to deconvolve bulk data of sEVs to the tissue of origin even to a certain cell type⁵⁻⁷, or characterize the number and cargo of these vesicles based on microfluidics⁸⁻¹⁰. However, a major limitation of these attempts is that they necessitate the isolation and purification of sEVs, which results in the loss of valuable information regarding the microenvironment of the original tissue. There is a pressing need for a methodology that can simultaneously capture the heterogeneity of both cells and sEVs. Moreover, there is no available technique capable of resolving the heterogeneity of sEVs at high-throughput and close to the level of individual sEVs without imposing additional requirements."

2) The authors need to clarify the advantages of their approach (profiling exosomes at

15

the droplet resolution). While several published studies have investigated single-cell extracellular vesicle secretion (see above), the current study is not really set up to measure specific vesicle secretion. Since it uses published scRNA-seq data, depending on the experimental preparation for these datasets, multiple unrelated vesicles could be encapsulated within a single droplet. Given the large size of the droplets with respect to exosome size, the study is essentially performing relatively bulk exosome profiling through droplet sampling. How is the approach advantageous as compared to dedicated single-cell vesicle secretion study as well as bulk exosome RNA profiling?

Response: Thanks for these insightful comments. We agree with the reviewer that our approach is to profile sEVs by droplet sampling. To highlight the advantages of our approach, we have summarized them into four key points. (1) SEVtras achieves a resolution in profiling sEVs that is exceptionally close to the level of individual sEVs, which enables a more detailed examination of the heterogeneity present within tissue-derived sEVs. (2) SEVtras maintains the connection between sEVs and their original cell types, allowing for the calculation of sEV secretion activity (ESAI) specific to certain cell types within complex tissue microenvironments. This introduces a new dimension in resolving and understanding cell states. (3) SEVtras offers a high-throughput capacity, enabling the characterization of thousands of sEVs in a single sample simultaneously. (4) SEVtras is easy to use and does not necessitate additional requirements or complex experimental setups. We have elaborated on these advantages in the last paragraph (Discussions) of the revised version.

3) The development of the EM framework and unified gene set is interesting but requires more rigorous validation. Currently, the experimental exosome-exclusive data was generated from total RNA seq (not from droplet-based sequencing). Additional studies are needed to answer key questions such as the minimal exosome counts per droplet required for accurate counting and classification, algorithm performance against complex background (other cell-free RNAs also encapsulated in the same droplets).

Response: We appreciate your suggestions. We have generated sEV-exclusive scRNA-seq data in both MSC and 293F cell lines and designed two types of noise spike-in experiments to address this concern. Please refer to **Question1** and **Question2** in the **Overall response**.

4) For exosome source tracking, the authors made several assumptions (1) the transcriptional profile of an exosome is more similar to its cell of origin and (2) exosome release is affected by the biogenesis capacity of the original cell type. How generalizable are these assumptions across datasets?

Response: The assumption 1 is the foundation of many sEV deconvolution methods (Shi et al., 2020, Science Advances) (Li et al., 2020, Computational and structural

biotechnology journal) (Zhu et al., 2021, PNAS). These methods used RNA profiles of the cell of origin to deconvolve sEVs, which endorses the assumption that the transcriptional profile of an sEV is more similar to its cell of origin. To provide further evidence, we performed a comparison of sEVs derived from MSC and 293F cell lines using scRNA-seq and found that sEVs from different cell types exhibit greater similarity to their respective cells of origin (**Fig. 2ab**). Please refer to **Question4** in the **Overall response**.

The assumption 2 is based on GSEA to reflect sEV biogenesis capacity of cells (Subramanian et al., 2005, PNAS). The idea of using gene set to represent sEV secretion has been supported by many studies. For example, in the breast cancer cells, sEV-associated gene set was identified to be enriched in 67NR cells in comparison with 4T1 cells, where 67NR generates more sEVs than the other cell type (Fathi et al., 2023, iScience). Moreover, we performed GSEA of sEV biogenesis capacity in MSC and 293F cells and found that the enrichment scores for MSC and 293F were 0.88 and -0.85, respectively. These findings are consistent with the results obtained from the ESAI analysis conducted on separate scRNA-seq experiments of MSC and 293F cell lines.

We have reworded text and citations in the revised version. Notably, we have changed “sEV source tracking” to “deconvolving sEV secretion activity” in the revised manuscript. By leveraging identified sEV-containing droplets, our method offers a higher level of precision compared to other approaches that solely rely on gene expression data at the bulk level.

5) Since multiple vesicles are likely to be encapsulated within a single droplet, for droplets that contain heterogeneous mixture of exosomes derived from different cellular origins, how were they classified?

Response: We appreciate the reviewer’s concern regarding the possibility of multiple types of vesicles being encapsulated within a single droplet. To test such possibility, we simulated the scenario of sEVs derived from different cellular origins. We conducted scRNA-seq experiments by mixing the cultured medium of MSC and 293F cells (**Fig. 2a-c**), and observed that the identified sEV droplets could be grouped into two distinct clusters. Notably, these clusters showed a high degree of overlap with those obtained from separate scRNA-seq experiments of MSC and 293F cells. Moreover, the ESAI measured in this mixed sample (MSC: 359%, 293F: 276%) was consistent with the values of the individual samples (MSC: 369%, 293F: 274%). Please refer to **Question4** in the **Overall response**. These findings provide support for the notion that ESAI could qualitatively reflect sEV secretion activity in MSC cells.

Therefore, we speculate that a majority of sEV-containing droplets likely contain sEVs originating from the same cell type. Firstly, in the mechanism of sEV biogenesis, when multivesicular bodies (MVBs) fuse with the plasma membrane, multiple sEVs

within the MVB are released by the cell at once. This suggests that sEVs from the same cell are likely to be present together. Secondly, sEVs from similar cells tend to aggregate into micro-sized clusters and localize in certain areas (Zomer et al., 2015, Cell). Consequently, these micro-sized clusters have a higher probability of being captured by a single droplet. Thirdly, SEVtras only classified droplets that surpass the detection threshold in the iterative algorithm as sEV-containing droplets.

6) How accurate is the approach as compared to other published tracking methods? Further investigation should be performed to assess the tracking accuracy, using pure vesicle populations derived from single cell source as well as known mixtures prepared with vesicles derived from different cells of origin.

Response: Thanks for the suggestion. As stated in our overall response, we designed scRNA-seq experiments on a mixture sample of MSC and 293F cell lines (Please refer to **Question4** in the **Overall response**). Based on this dataset, we compared our deconvolution method with two state-of-the-art methods: CIBERSORT (Newman et al., 2019, Nature biotechnology) and EV-origin (Li et al., 2020, Computational and structural biotechnology journal). As shown in **Extended Data Fig. 4b**, our method closely approximates the real composition of sEVs origin.

Extended Data Fig. 4b Comparisons of sEV composition deconvolution among ESAL_c, CIBERSORT and EV-origin. scRNA-seq data of MSC and 293F mixtures was used as the benchmark dataset.

7) Induction and inhibition of exosome secretion could be included in multiple cell types to evaluate the responsiveness of ESAL.

Response: Thanks for this valuable suggestion. As suggested, we treated cultured MSC cells with sEVs secretion inducer: monensin sodium salt. Moreover, we also spiked sEVs into MSC single-cell suspension. Please refer to **Question3** in the **Overall response**.

Reviewer #2:

Remarks to the Author:

The authors present exoTras, a workflow that identifies exosomes containing droplets from single cell RNA-seq datasets. The algorithm is unique, novel and much needed to the field of extracellular vesicles (EVs). Though this reviewer is supportive of this article to be published, certain parts of the manuscript need to be updated.

Response: We thank the reviewer for supporting the novelty and significance of SEVtras. We have thoroughly revised our manuscript and added more experiments to evaluate the performance of SEVtras.

Each section needs more clarity as what was done and what is reported in the figures. Now it is quite vague and hard to understand.

Response: Thanks for the suggestion. We have made extensive alterations to the structure, format, presentation and analysis of our manuscript.

1. The authors need to add additional data on the characterization of exosomes/EVs from single cell seq datasets and their workflow. What controls are in place to ensure that these are indeed EVs and not cell debris? Perhaps the authors can compare cells, debris (intentionally obtained with harsh treatment or stress), EVs for EV markers (CD63, CD81, Alix, Sytenin, TSG101) and negative (GRP96, Calnexin). This can be done on A549 cell dataset (Figure 1c) as well. The cells can be broken into debris by passing them through sieve or mechanical stress and compared with EVs and cell lysates.

Response: Thank you for the suggestion. We have performed bulk RNA-seq experiments to compare cells, debris, large EVs and sEVs in both MSC and 293F cell lines. Furthermore, we have spiked cell debris and large EVs into scRNA-seq samples. Please refer to **Question2** in the **Overall response**.

2. What quality control is placed to check that the sample used does not suffer from damage? Is there a possibility of including a negative control? Perhaps a RNA profile signature from debris?

Response: We have two types of quality control (QC) to ensure sample quality. The first one is that the mapping rate should be higher than 90%. This QC is to ensure that the samples are not affected by exogenous contamination and over fragmentation. The second is that the percentage of mitochondrial gene counts should be less than 15%. This QC is to remove droplets associated with mitochondria and lysing cells. We have added additional text to the Methods of the revised manuscript.

3. ESAI is very interesting and informative. Though this is very useful, what if the EVs are migrating from adjacent cells? How does the algorithm tackle this? Hence the ESAI should also consider the cell type source for EVs and their locality in the tissue source of interest.

Response: We thank the reviewer for this insightful suggestion. To avoid counting sEVs that migrate from the adjacent cells, we have set a limitation in the process of ESAI deconvolution. Specifically, if a given sEV is not similar to any cell types, we do not consider it in the later calculation. Furthermore, we have discussed the limitation of SEVtras in solving migrating sEVs in the last paragraph (Discussions) of the revised manuscript.

4. Throughout the manuscript it is not clear what data is generated by the authors and those that are utilised from publicly available resources. Perhaps the authors can clarify in the manuscript. This also brings a point that the authors need to do this comprehensively on one dataset that they generated inhouse – where all characterization such as western blot for markers, quality control, debris all needs to be done. The existence of two exosomes group can also be shown comprehensively. The authors can apply SEVtras to other datasets later like in the manuscript and shows its utility.

Response: Thanks for pointing this out. We have clarified the public and our dataset in the revised version. And we have ensured that the experiments that we have done are comprehensive and include all characterizations (Please refer to **Supplementary Figure 2** and **Overall response**).

5. What does ESAI of 8.3% mean? Does it mean X of exosomes droplets / Y of total cells? How does the authors deal with EV subtypes and heterogeneity here? What about non-exosomes droplets but not cell droplets? The manuscript lacks clarity at various levels for every section.

Response: ESAI at the sample/tissue level is defined as the number of sEV-containing droplets / the number of cells.

ESAI at the cell type level is the number of sEV-containing droplets secreted by a certain cell type / the number of cells in a certain cell type.

To distinguish these two levels, the ESAI at the cell type level is referred to as “ESAI_c”. We applied ESAI_c to characterize the heterogeneity of cells/sEV type. Non-sEV droplets and non-cell-containing droplets will not influence the result of ESAI as they do not occur in the equations. Please refer to **Question2** in the **Overall response**. We have reworded the description in the revised version.

6. ESAI in figure 2e. How was this done? The legend is not clear? What does the error bar mean? Why is the error bar for Tumor adjacent not equal on both sides?

Response: We are sorry for this misunderstanding. We used the “barplot” function of the seaborn package in Python to plot Figure 2e (Fig. 2h in the revised version). The bar represents the mean, and the error bar is the 95% confidence interval with 1,000 bootstrap iterations. Hence, the lower and upper limits in the error bar are the values corresponding to the 2.5th and 97.5th percentiles after bootstrapping. We have clarified this in the legend of the revised version.

7. The use of the term “exosome” need to be avoided throughout the manuscript including figures. Exosome refer to the RNA complex while exosomes (plural) refers to endocytic driven extracellular vesicles.

Response: We have avoided the term “exosome” in the revised version.

8. The authors should use the term extracellular vesicles (EVs) or small EVs in general as opposed to exosomes. Exosomes are endocytically derived sEVs but no current technology can purify exosomes alone to 100% purity.

Response: We agree with the reviewer that we are currently unable to purify exosomes alone. We have used the term “sEVs” to replace “exosomes” in the revised version. And we have changed the name of our method from “exoTras” to “SEVtras”.

Reviewer #3:

Remarks to the Author:

The manuscript by He et al proposes a new algorithm (‘exoTras’) via which the RNA content and potential tissue origin of ‘exosomes’ (a subtype of extracellular vesicles) may be assessed. Hereto, the authors re-analyze droplet-based single cell RNAseq datasets. Droplet sequencing of (single) extracellular vesicles (EVs) isolated from culture medium or bodyfluids has been performed previously. However, these previous approaches did not allow tracing back the origin of the EVs to their secretory cell types.

Response: We thank the reviewer for appreciating the novelty and significance of our study.

The new approach proposed here is based on the idea that libraries of droplet-based scRNAseq experiments are composed of cell-containing droplets, vacant droplets, cellular debris-containing droplets and exosome-containing droplets. It took me some

time to understand the conceptual background of this approach, but I presume that it is based on the idea that droplets that contain RNA but no cells are thought to contain extracellular RNA (e.g. enclosed in EVs) released by the cells in question. I have doubts on the foundation of this approach based on the following arguments:

Response: We have added extensive control experiments to validate the performance of SEVtras in the revised version.

1. It is highly unlikely that the material that is found in non-cell-containing droplets of the scRNAseq experiments are actively released extracellular vesicles. The cells (either from cell cultures or detached from tissues by enzymatic treatments) are processed by washing and pelleting, and in some cases were subsequently cryopreserved. There was no recovery / culture step just prior to entering cells in the scRNAseq system (e.g. Chromium system). Therefore there has been no time/good condition for cells to actively release natural EVs. It is therefore far more likely that the extracellular RNAs are from rip-offs, which indeed would reflect the RNAome of the cells that had been processed.

Response: We appreciate the reviewer's concern regarding the potential removal of natural sEVs during the pre-processing steps of droplet-based scRNA-seq experiments. To address this concern, we firstly measured the concentration of sEVs in three types of tissues using nanoparticle tracking analysis and found a considerable amount of sEVs in fresh tissues (**Extended Data Fig. 1ab**). To investigate whether the scRNA-seq pre-processing steps remove a substantial amount of sEVs, we utilized NanoLuc-labelled sEVs, which exhibit luminescence upon the addition of specific substances. By measuring the luminescence intensity, we could assess the concentration of sEVs in the samples before and after the scRNA-seq pre-processing steps (**Extended Data Fig. 1c-e**). Specifically, equal amounts of labelled sEVs were spiked into two sample groups: one group underwent standard scRNA-seq pre-processing, while the other served as a control without any treatment. Notably, we observed a significant amount of labelled sEVs remaining after the scRNA-seq pre-processing steps. These findings indicate that a considerable proportion of sEVs could be retained and captured during the scRNA-seq process, subsequently allowing their identification using SEVtras.

We have added these results as **Extended Data Fig. 1** in the revised manuscript.

Extended Data Fig. 1. Presence of sEVs in the scRNA-seq samples. **a**, Schema of nanoparticle tracking analysis (NTA) for fresh tissues. **b**, Size distribution of sEVs in the three types of mouse tissues. From left to right: brain, spleen and kidney. **c**, Luminescence intensity of NanoLuc-labelled sEVs at variable concentration. The luminescence was measured by microplate reader (see Methods). **d**, Luminescence intensity of background. **e**, Changes in luminescence after scRNA-seq pre-processing steps. Luminescence was measured 10 minutes after adding the Nano-Glo® Luciferase Assay Reagent.

In relation to the previous point, it is very difficult to substantiate that the observed differences in the amount of ‘exosome’ material between different (parts of) tissues is really indicative of differences in the amount of EVs these tissue cells release (indicated by the authors as the ESAI: ‘exosome secretion activity index’). The differences may just be a consequence of differences in the fragility of different cell types or the difficulty with which each cell type can be freed from the tissue environment.

Response: We have designed additional experiments to evaluate the reliability and accuracy of ESAI. Please refer to **Question3** and **Question4** in the **Overall response**.

2. The authors claim to be able to differentiate between ‘debris’ and ‘exosomes’. First, it is currently not possible to differentiate between endosome-derived EVs (exosomes) and other types of actively released EVs once they are present outside cells. There is no evidence in this manuscript that the observed material is from genuine exosomes. Second, the adopted approach to differentiate between ‘debris’ and ‘exosomes’ is not

sound. The reference EV transcriptome has been composed by manually curating EV-derived RNA gene sets from online available databases (exoCarta, ExRNA atlas). Especially the exoCarta database contains data of studies where only an enrichment step has been performed to isolate EVs. These samples are highly contaminated with non-EV-associated RNAs and therefore not suitable to use as specific reference for EV (or exosome)-associated RNAs. This, in combination with the previous point, severely complicates definitions of 'exosomes' versus 'debris'.

Response: Thank you for these comments. On the one hand, we first performed RNA-seq analysis of sEVs, debris and large EVs, and observed distinct transcriptional profiles between sEVs and the other components (debris and large EVs) in both MSC and 293F samples. To further investigate the potential impact of debris and large EVs on our method, we spiked these components into our samples and performed scRNA-seq experiments. We found that ESAI was able to tolerate the presence of cell debris and large EVs, suggesting that our method is robust and can effectively distinguish sEVs from other components. Please refer to **Question2** for details in the **Overall response**.

While we acknowledge that the databases we utilized for SEVtras contain inherent noise, we have taken steps to address this issue. Specifically, we have developed an iterative approach using Expectation-Maximization algorithm to enhance the reliability of sEV identification. In this revised version, we have also discussed the importance of high-quality reference databases for sEVs.

3. It is unclear how is assessed whether there is one or multiple EVs in a drop and the interpretation thereof (e.g. are these multiple EVs from one or multiple cell types?).

Response: We appreciate the reviewer's concern regarding the possibility of multiple types of vesicles being encapsulated within a single droplet. To test such possibility, we simulated the scenario of sEVs derived from different cellular origins. We conducted scRNA-seq experiments by mixing the cultured medium of MSC and 293F cells (**Fig. 2a-c**), and observed that the identified sEV droplets could be grouped into two distinct clusters. Notably, these clusters showed a high degree of overlap with those obtained from separate scRNA-seq experiments of MSC and 293F cells. Moreover, the ESAI measured in this mixed sample (MSC: 359%, 293F: 276%) was consistent with the values of the individual samples (MSC: 369%, 293F: 274%). Please refer to **Question4** in the **Overall response**. These findings provide support for the notion that ESAI could qualitatively reflect sEV secretion activity in MSC cells.

Therefore, we speculate that a majority of sEV-containing droplets likely contain sEVs originating from the same cell type. Firstly, in the mechanism of sEV biogenesis, when multivesicular bodies (MVBs) fuse with the plasma membrane, multiple sEVs within the MVB are released by the cell at once. This suggests that sEVs from the same

cell are likely to be present together. Secondly, sEVs from similar cells tend to aggregate into micro-sized clusters and localize in certain areas (Zomer et al., 2015, Cell). Consequently, these micro-sized clusters have a higher probability of being captured by a single droplet. Thirdly, SEVtras only classified droplets that surpass the detection threshold in the iterative algorithm as sEV-containing droplets.

4. With regard to the previous arguments, the manuscript would have largely benefited from including solid experimental validation. Now, most of the 'validations' are simulations, which are not sufficient for substantiating the credibility of this entirely new approach. Experimental validation for assessing whether actively released EVs are measured and for the ESAI could for example be a droplet scRNAseq analysis of cultured cells treated or not with bafilomycin A1 as a reagent known to induce increased EV release. In addition, the capacity of exoTras to distinguish EV- versus debris-associated RNA could be validated by performing droplet scRNAseq of cells spiked with variable ratios of highly purified EVs and mechanically desintegrated cells (i.e. 'debris').

Response: Thanks for this suggestion. We have performed scRNA-seq of sEVs in MSC and 293F cell lines to validate the performance of SEVtras. Please refer to **Question1** for details in the **Overall response**. We also treated cultured MSC cells with sEV secretion inducer, monensin sodium salt, and performed scRNA-seq. To demonstrate them more clearly, we also spiked sEVs in single cell suspension and performed scRNA-seq. Please refer to **Question3** for details in the **Overall response**. Furthermore, we have spiked cell debris and large EVs in scRNA-seq samples, and performed scRNA-seq. Please refer to **Question2** for details in the **Overall response**.

Additional comment:

The authors should clarify in more detail what the application area of the ESAI and 'EV' typing is: do they expect that this SEVtras method can be used for diagnosis of cancer, and which input material would be required for this? Is it expected that SEVtras data reveal disease-associated exRNA transcriptomes that might be traced back in body fluids used as liquid biopsies?

Response: We believe that SEVtras will contribute to both basic discovery and clinic applications in the field of EVs. The versatility of SEVtras enables its potential applications in a range of research areas. For instance, SEVtras can be utilized to study the crosstalk between different cell types in complex tissues and decipher their sEV-mediated interactions. sEVs are actively involved in tumor migration and invasion, our method can shed light on underlying regulatory mechanisms in tissue microenvironment. Furthermore, SEVtras is able to identify cell types with altered sEV secretion activity, which may serve as a potential indicator for tumor progression from

benign to malignant.

Reviewer #4:

Remarks to the Author:

Exosomes are extracellular vesicles involved in the intercellular transportation of materials including nucleic acids, proteins, and/or other regulatory factors. They play vital roles in cell-cell communications regulating development, immune response, and disease onset and progression, and have been engineered as a vehicle for drug delivery. To elucidate the heterogeneity of exosome cargos, to identify their source of origin, and to unravel the regulatory roles of different types of exosomes have attracted much research.

The study by He et. al very nicely mounted to the ample availability of droplet-based single cell sequence data to develop an analytic tool set, exoTras, for exosome analysis. The main contribution includes a few algorithms to identify exosome containing droplets, to estimate exosome secretion activity for different types of cells, and to track back to their secretory cell types. The study also applied SEVtras to analyze the droplet scRNA-seq data for normal tissues and cancer samples, and found exosome subtypes of different cargo type and cell source, as well as the elevated exosome secretion activity of aggressive epithelial cells in CRC and the potential usage of exosomes in biomarker for vascular invasion in early-stage cancer development.

This SEVtras tool could be a powerful one that can dig into huge amount existing data from of droplet-based scRNA-seq studies (and requires no additional invest in new studies).

Response: We greatly appreciate the reviewer's comments on the novelty and significance of our method.

The following are my comments:

1, it is not clear from the description how the authors deal with droplets of mixed content of cell debris and exosomes. I guess the authors know this, but the writing gives no clue on how these are handled and how it will affect the performance of exoTras.

Response: We have re-organized the part of the cell-free droplets simulation to make it easy to follow. In addition, we have spiked cell debris into single-cell suspension and performed scRNA-seq to validate the performance of SEVtras. Please refer to **Question2** for details in the **Overall response**.

2, the normalization of different samples is not well-described in the manuscript. How is the sequencing depth handled as a factor? I have a concern looking at Figure 1b--

26

apparently total UMI affects the AUC. What about the other types of confounding factors considered? Some sample preparation protocols may deplete exosomes and the others may enrich them. Some experimental settings may favor a droplet to contain more exosomes and the others may disfavor it. The sequencing platform may favor the obtaining of specific exosome content. All of these factors may affect the sequencing results; it is very much like batch effects in single-cell data analysis. Without careful normalization and batch effect removal, it is dangerous to compare across samples.

This is especially a factor that needs to be carefully taken care of for the cancer applications in page 7.

Response: We agree with reviewer that sequencing depth and other confounding factors may affect the downstream analysis of SEVtras. Hence, we first collected the information of sequencing depth and other related confounding factors of all samples we used in the **Supplementary Table S3**. We removed the sample with low saturation in the downstream analyses. To eliminate the influence of confounding factors, we further analyzed the data of different studies separately. When it is necessary to compare data from different studies together, we used a linear regression model to eliminate confounding factors in related analyses. Third, we have also added a more detailed description on the normalization methods in the Methods section of the revised manuscript.

3, in the simulation study, the cell-exclusive sample are likely to be associated with exosomes as well. Please be precise. Also, will this be a problem? See above, what if the fraction of associated exosomes varies? Will this be a confounding factor?

Response: Yes, the transcriptome profile of sEVs is expected to be similar to their original cells to some extent. While the cargo of sEVs may be randomly loaded from cells, specific protein sorting mechanisms are involved in sEV biogenesis and content loading. These mechanisms contribute to the enrichment of distinct proteins and nucleic acids in sEVs compared to their parent cells. SEVtras considers the differences in UMI counts between sEVs and cells, ensuring that this disparity does not pose a problem for the analysis. The focus of SEVtras is to identify the unique components specific to sEVs, which are less likely to be influenced by other cellular noises.

To demonstrate our statement, we performed RNA-seq in cell, debris, large EVs and sEVs in both MSC and 293F cell lines. We found that transcriptional profiles of sEVs are distinct to those of other components. Please refer to **Question2** for details in the **Overall response**. We further spiked sEVs into scRNA-seq samples and calculated ESAI. As expected, we found that ESAI can accurately decipher the sEV secretion activity of cells. Please refer to **Question3** for details in the **Overall response**.

4, figure 1c, the correlation coefficients seem low even for droplets. This is rather

expected given the little amount of material in each droplet and the data loss in scRNA-seq. Would exoTras benefit from data imputation?

Response: Thanks for the suggestion. We totally agree with the reviewer that imputation would be helpful for SEVtras. We have incorporated the use of the data imputation method, MAGIC (Van Dijk et al., 2018, Cell), to evaluate the improvements (**Extended Data Fig. 3b**). We found that the correlation between the sEVs identified by SEVtras and experimentally isolated sEVs improved after applying the imputation method in both the MSC and 293F cell lines. We have added the imputation function to SEVtras.

Extended Data Fig. 3b. sEVs identified by SEVtras were significantly correlated with experimentally isolated sEVs in both MSC and 293F cell lines. Here, we added up RNA counts of experimentally isolated sEVs in the ground truth data as pseudo-bulk sample for comparison. Left panel: Pearson correlation between droplets identified by SEVtras and the pseudo-bulk data in MSC and 293F cell lines. Droplets were classified into two classes: sEV-containing (yellow) and debris-containing (grey) in the boxplot. Right panel: matrixes of debris and sEVs were imputed by MAGIC (Van Dijk et al., 2018, Cell).

5, the study only investigated A549 cells. It would be very nice if the authors can perform more cell lines and to compare. It will also help the cell source tracing (for example, being a validation).

Response: We have applied scRNA-seq on MSC and 293F cell lines. Please refer to **Question1** in the **Overall response**. We also mixed the two cell lines at 1:1 ratio to validate the performance of ESAI for different cell types. Please refer to **Question4** in the **Overall response**.

6, citation to supplementary figures should also be clear onto panels.

Response: We have revised them accordingly.

Minor

1, page 3, middle paragraph, I do not agree that the decrease of specificity is slight.

Response: We have re-worded this part in the revised version.

2, page 5~6, the data are from scRNA-seq studies of different tissues/organs, and the analyses are on constituent cell types. But the writing is a little confusing without the introduction of cell types. I made a mistake at the first time thinking that the authors confuse tissues and cell types. Better to mention the two to be less ambiguous.

Response: As suggested, we have revised the manuscript accordingly.

Decision Letter, first revision:

Dear Fangqing,

Thank you for submitting your revised manuscript "SEVtras delineates small extracellular vesicles at droplet resolution from single-cell transcriptomes" (NMETH-BC50796B). It has now been seen by the original referees and their comments are below. The reviewers find that the paper has improved in revision, and therefore we'll be happy in principle to publish it in Nature Methods, pending minor revisions to satisfy the referees' final requests and to comply with our editorial and formatting guidelines.

I strongly recommend addressing all the remaining concerns raised by Ref 3.

TRANSPARENT PEER REVIEW

Nature Methods offers a transparent peer review option for new original research manuscripts submitted from 17th February 2021. We encourage increased transparency in peer review by publishing the reviewer comments, author rebuttal letters and editorial decision letters if the authors agree. Such peer review material is made available as a supplementary peer review file. Please state in the cover letter 'I wish to participate in transparent peer review' if you want to opt in, or 'I do not wish to participate in transparent peer review' if you don't. Failure to state your preference will result in delays in accepting your manuscript for publication.

ORCID

IMPORTANT: Non-corresponding authors do not have to link their ORCIDs but are encouraged to do so. Please note that it will not be possible to add/modify ORCIDs at proof. Thus, please let your co-authors know that if they wish to have their ORCID added to the paper they must follow the procedure

described in the following link prior to acceptance:

Sincerely,
Madhura

Madhura Mukhopadhyay, PhD
Senior Editor
Nature Methods

Reviewer #2 (Remarks to the Author):

The authors have addressed the concerns.

Reviewer #3 (Remarks to the Author):

The revised manuscript by He et al describes the development and testing of an algorithm ('SEVtras') via which the RNA content and potential tissue origin of EVs may be assessed. Besides analysis of published droplet-based single cell RNAseq datasets, the authors also included experiments in which they performed sc-RNAseq experiments themselves to validate their conclusions. Compared to the original submission, this revised version has substantially improved. A large number of reviewer questions has been addressed. Yet, the rationale behind some of the experimental strategies is difficult to understand. In general, the manuscript also suffers from a large degree of complexity, lack of described experimental details, and figures being scattered over 3 different locations (main, extended data sets, and suppl figures).

Main comments:

General:

Several of the figure panels (especially those in suppl and extended data figures) are not described in the main text. On top of that, there is continuous switching between main figures, extended data figures and supplementals. All of this makes it extremely difficult to follow and understand the manuscript.

Feasibility of identifying sEV-containing droplets from scRNA-seq:

- From a biology-perspective it is questionable whether cells that are being processed for sc-RNAseq analysis have sufficient time and are in good condition to actively release EVs as they would do in an in vivo or in vitro culture condition. Although this is a difficult question to answer, it is important to report exactly each step from harvesting of the tissue to dissociation of cells, to processing for sc-RNAseq analysis. I cannot find ANY information about this in the manuscript!
- In Extended Data fig 1a-b the authors process tissues and look at EVs, but it is unclear how these experiments have been performed (no descriptions in Methods section!) and which question this answers. Are these the EVs that are present in the tissue prior to making single cell suspensions? If cells are washed during these procedures, all EVs present in tissues at the time of isolation will be discarded.... How is this data relevant?
- The authors aim to address the question above in Extended Data fig 1 c-e, but instead of assessing the processing steps to isolate the cells, the authors assessed whether the processing steps affected recovery of spiked-in EVs. That is not relevant, because the EVs present in the tissue at the time of harvesting are not the same as those that are released by cells at the timepoint of sc-RNAseq analysis.
- The methodology used in extended data fig 1 c-e is expression of nano-luc in cells and analyzing nano-luc signals in supernatants. However, this nano-luc can be released by cells in EV-associated and non-EV-associated forms, and is therefore not indicative for EV release. To prevent this, the nano-luc should be coupled to proteins frequently sorted into EVs (eg tetraspanins, as published before by several groups).

Performance evaluation of SEVtras:

In figure 1b-c the authors included wet lab experiments to validate their technique. Yet, the selected approaches raise several questions.

- The monensin treatment protocol should be added to the Methods section
- The Methods section should also contain a detailed section on how the sc-RNAseq experiments were performed: how were the cells harvested, how many cells were processed, which washing steps were performed, how were libraries prepared, etc.
- The Methods section should also report to which number of MSC the EVs from 1e4 MSC were added to do the spike-in sc-RNAseq evaluation experiments of figure 1.
- It is unclear and confusing why the authors compare +Mon & 1x sEV to + 3x sEVs. The authors should also show the + 1x sEVs data. If the method is insensitive in the low range numbers of (spiked) EVs, this should be reported
- The CITE-seq analyses should be better described. What are the 'sEV marker-positive droplets'? Are these droplets without cells but with (a certain amount of?) CD9 and/or CD63 protein signal?

Deconvolution of SEVtras:

- Methods for the experiments shown in figure 2a-c should be described in full experimental detail in the Methods section
- Why do the EV plots in figure 2b right panel look different from the ones in figure 2a right panel? What does this say about the methodology or the biology?

Minor:

The authors have now changed the term 'exosomes' into small EVs (sEVs). Yet, they should explain how they define this subset in their studies.

- The authors define the 2000 x g EVs as large EVs. However, EVs sedimenting at 10,000 x g are generally also defined as large EVs. Why did the authors not analyze this subset? It is very likely a subset that occurs in the sc-RNAseq analyses.
- Lane 83: the used databases are not specifically for small EVs (if these are defined as the EVs that sediment at 100,000 x g).

Reviewer #4 (Remarks to the Author):

The revisions have effectively addressed my major concerns regarding the study. I am pleased to see that the imputation has improved the quality of the data, and the investigation of multiple cell lines has demonstrated SEVtras' ability to deconvolute different cell sources.

Although I have only a few minor suggestions:

1. It may be helpful to provide an overall statistic on sEV-containing droplets from the SEVtras analysis for each scRNA-seq dataset examined in this study.
2. In line 79, could you please clarify how droplets containing both cell debris and sEVs are handled?
3. In Fig 1b, it is not clear which dots represent the SEVtras analysis and which ones represent experimentally resolved debris.

Overall, I believe that SEVtras could be an exceptionally powerful tool for extracting biological insights and discoveries from the ever-increasing scRNA-seq studies, and may promote dedicated investigations, including potential biomedical applications. Just curiosity, I am not sure on the deconvoluting capacity of

ESAI scores. Since they are numerical values, would it be a problem for deconvoluting complex tissues? Does it make more sense to use a characteristic vector for deconvolution?

Author Rebuttal, first revision:

Dear Madhura,

Thank you very much for considering our paper and providing us with the opportunity in principle to publish it in *Nature Methods*. We have addressed all of the reviewers' comments and very much appreciate the valuable suggestions from both the editorial and reviewers. The responses to these comments are attached as follows.

Best regards,

Fangqing

Reviewer #2 (Remarks to the Author):

The authors have addressed the concerns.

Response: We thank Reviewer #2 for their previous suggestions, which greatly improves our manuscript.

Reviewer #3 (Remarks to the Author):

The revised manuscript by He et al describes the development and testing of an algorithm ('SEVtras') via which the RNA content and potential tissue origin of EVs may be assessed. Besides analysis of published droplet-based single cell RNAseq datasets, the authors also included experiments in which they performed sc-RNAseq experiments themselves to validate their conclusions. Compared to the original submission, this revised version has substantially improved. A large number of reviewer questions has been addressed. Yet, the rationale behind some of the experimental strategies is difficult to understand. In general, the manuscript also suffers from a large degree of complexity, lack of described experimental details, and figures being scattered over 3 different locations (main, extended data sets, and suppl figures).

Response: We thank Reviewer #3 for the insightful comments and appreciation for our improvements. We have provided a clearer description of the experimental details and restructured the supplementary figures to reduce the complexity of our paper.

Main

comments:

General: Several of the figure panels (especially those in suppl and extended data figures) are not described in the main text. On top of that, there is continuous switching between main figures, extended data figures and supplementals. All of this makes it extremely difficult to follow and understand the manuscript.

Response: Thanks. We have restructured the supplementary figures and merged them into extended data figures.

Feasibility of identifying sEV-containing droplets from scRNA-seq:

- From a biology-perspective it is questionable whether cells that are being processed for sc-RNAseq analysis have sufficient time and are in good condition to actively release EVs as they would do in an in vivo or in vitro culture condition. Although this is a difficult question to answer, it is important to report exactly each step from harvesting of the tissue to dissociation of cells, to processing for sc-RNAseq analysis. I cannot find ANY information about this in the manuscript!

Response: Thank you. We have added a detailed protocol for scRNA-seq processing in the Methods section “Single cell RNA-seq processing”.

- In Extended Data fig 1a-b the authors process tissues and look at EVs, but it is unclear how these experiments have been performed (no descriptions in Methods section!) and which question this answers. Are these the EVs that are present in the tissue prior to making single cell suspensions? If cells are washed during these procedures, all EVs present in tissues at the time of isolation will be discarded.... How is this data relevant?

Response: Thanks for pointing this out. We have added details of sEVs separation in the Methods section “Tissue sEV separation and NTA analysis”. Based on the results in Extended Data Fig. 1ab, we speculate that a large number of sEVs are present in the tissue prior to making single cell suspension. To further assess the influence of scRNA-seq procedures (e.g. cell washing) on the retention of existing sEVs, we found that a substantial proportion of sEVs would be retained in the scRNA-seq sample (Please refer to Fig. 1a and Extended Data Fig. 1ef).

- The authors aim to address the question above in Extended Data fig 1 c-e, but instead of assessing the processing steps to isolate the cells, the authors assessed whether the processing steps affected recovery of spiked-in EVs. That is not relevant, because the EVs present in the tissue at the time of harvesting are not the same as those that are released by cells at the timepoint of sc-RNAseq analysis.

Response: Extended Data Fig. 1ab showed a large number of sEVs are present in the tissue. Due to scRNA-seq sample preparation requires to be very fast to ensure cell viability, there was no time for cells to secrete enough sEVs. Hence, existing sEVs will contribute to the majority of sEVs in the scRNA-seq sample. To more clearly track the retention of these previously existing sEVs during scRNA-seq processing, we spiked NanoLuc-labelled sEVs to represent them. We found that a substantial proportion of those sEVs would be retained in the scRNA-seq sample (Fig. 1a and Extended Data Fig. 1ef).

- The methodology used in extended data fig 1 c-e is expression of nano-luc in cells and analyzing nano-luc signals in supernatants. However, this nano-luc can be released by cells in EV-associated and non-EV-associated forms, and is therefore not indicative for EV release. To prevent this, the nano-luc should be coupled to proteins frequently sorted into EVs (eg tetraspanins, as published before by several groups).

Response: Actually, we have conducted sequential differential ultracentrifuge procedures to isolate Nanoluc-labelled sEVs. We have stated more details of Nano-Luc labelled sEVs in the Methods section “NanoLuc-labelled sEVs preparation”.

To verify the specificity of NanoLuc protein in sEVs, we first performed NTA analysis. As shown in Extended Data Fig. 1c, the size distribution of NanoLuc-labelled sEVs matched the range of sEVs. To further eliminate the effect of non-sEV-associated forms, we added Triton X and Proteinase K treatments to these sEVs and evaluated NanoLuc luminescence. We found no change in luminescence when NanoLuc-labelled sEVs were treated with Triton X-100 or Proteinase K alone (Extended Data Fig. 1d). The luminescence disappeared only when treated with both Triton X-100 and Proteinase K, suggesting that NanoLuc protein is specifically localized in sEVs.

Performance evaluation of SEVtras:
In figure 1b-c the authors included wet lab experiments to validate their technique. Yet, the selected approaches raise several questions.

- The monensin treatment protocol should be added to the Methods section

Response: Thank you for this suggestion. We have added details in the Methods section “sEV secretion activity stimulation”.

- The Methods section should also contain a detailed section on how the sc-RNAseq experiments were performed: how were the cells harvested, how many cells were processed, which washing steps were performed, how were libraries prepared, etc.

Response: Thanks. We have added details in the Methods section “Single cell RNA-seq processing”.

- The Methods section should also report to which number of MSC the EVs from 1e4 MSC were added to do the spike-in sc-RNAseq evaluation experiments of figure 1.

Response: Thanks for the suggestion. To ensure consistency, all samples in scRNA-seq were set as 7,000 cells, including the mixture sample of MSC and 293F (3,500 MSCs + 3,500 293F cells).

- It is unclear and confusing why the authors compare +Mon & 1x sEV to + 3x sEVs. The authors should also show the + 1x sEVs data. If the method is insensitive in the low range numbers of (spiked) EVs, this should be reported

Response: To assess the impact of ESAI with low range number of sEVs, we compared the sample of “+Mon & 1x sEVs” to “+ Mon alone”. SEVtras was able to detect an increase in ESAI between the two samples. In addition, we found that the increase was about 1/3 of the increase in the “+ 3x sEVs” treatment, indicating that SEVtras is reliable in the sEV secretion activity measurement. We have reworded the related part for better understanding.

- The CITE-seq analyses should be better described. What are the ‘sEV marker-positive droplets’? Are these droplets without cells but with (a certain amount of?) CD9 and/or CD63 protein signal?

Response: We have described it more clearly in the revised manuscript. The “sEV marker-positive droplets” are droplets that are positive for either CD9 or CD63.

Deconvolution of SEVtras:

- Methods for the experiments shown in figure 2a-c should be described in full experimental detail in the Methods section

Response: We have added experimental details in the Methods section “Single cell RNA-seq processing”.

- Why do the EV plots in figure 2b right panel look different from the ones in figure 2a right panel? What does this say about the methodology or the biology?

Response: Thanks. The difference may relate to cell states because these cells of the sample comes from different plate. After BBKNN batch adjustment with default parameters (Polanski et al. 2019), we found that sEVs from the two kinds of samples showed similar distribution. Please refer to Extended Data Fig. 6b.

Minor:

The authors have now changed the term ‘exosomes’ into small EVs (sEVs). Yet, they should explain how they define this subset in their studies.

Response: We have added the definition in the revised manuscript. Our study focused on sEVs enriched by sequential differential ultracentrifuge procedures (see Methods).

- The authors define the $2000 \times g$ EVs as large EVs. However, EVs sedimenting at $10,000 \times g$ are generally also defined as large EVs. Why did the authors not analyze this subset? It is very likely a subset that occurs in the sc-RNAseq analyses.

Response: Thanks for pointing this out. As recommended by the MISEV2018 guidelines (Théry et al, 2018), we define large EVs as diameter larger than 200 nm. Whole-mounts of materials recovered in the $2,000 \times g$ pellet showed vesicles in majority larger than 200 nm in diameter (Kowal et al, 2016). In contrast, most vesicles pelleting at $10,000 \times g$ were smaller than 200 nm in size. To minimize the influence of impurity, we chose $2,000 \times g$ pellet for further comparisons.

- Lane 83: the used databases are not specifically for small EVs (if these are defined as the EVs that sediment at $100,000 \times g$).

Response: We have revised our manuscript to state that we have manually curated genes that are associated with sEVs.

Reviewer #4 (Remarks to the Author):

The revisions have effectively addressed my major concerns regarding the study. I am pleased to see that the imputation has improved the quality of the data, and the investigation of multiple cell lines has demonstrated SEVtras' ability to deconvolute different cell sources. Although I have only a few minor suggestions:

1. It may be helpful to provide an overall statistic on sEV-containing droplets from the SEVtras analysis for each scRNA-seq dataset examined in this study.

Response: Thanks for the suggestion. We have added these statistics in Supplementary Table S3.

2. In line 79, could you please clarify how droplets containing both cell debris and sEVs are handled?

Response: Thanks for pointing this out. We have modified our manuscript to clarify that we have set a threshold for the sEV signal score.

3. In Fig 1b, it is not clear which dots represent the SEVtras analysis and which ones represent experimentally resolved debris.

Response: As suggested, we have added a detailed description.

Overall, I believe that SEVtras could be an exceptionally powerful tool for extracting biological insights and discoveries from the ever-increasing scRNA-seq studies, and may promote dedicated investigations, including potential biomedical applications. Just curiosity, I am not sure on the deconvoluting capacity of ESAI scores. Since they are numerical values, would it be a problem for deconvoluting complex tissues? Does it make more sense to use a characteristic vector for deconvolution?

Response: We appreciate the reviewer for these supportive comments. SEVtras has provided customized parameters to achieve this.

Final Decision Letter:

Dear Fangqing,

I am pleased to inform you that your Article, "SEVtras delineates small extracellular vesicles at droplet resolution from single-cell transcriptomes", has now been accepted for publication in Nature Methods. Your paper is tentatively scheduled for publication in our * print issue, and will be published online prior to that. The received and accepted dates will be 1st Nov, 2022 and 30th Oct, 2023. This note is intended to let you know what to expect from us over the next month or so, and to let you know where to address any further questions.

Over the next few weeks, your paper will be copyedited to ensure that it conforms to Nature Methods style. Once your paper is typeset, you will receive an email with a link to choose the appropriate publishing options for your paper and our Author Services team will be in touch regarding any additional information that may be required.

You will receive a link to your electronic proof via email with a request to make any corrections within 48 hours. If, when you receive your proof, you cannot meet this deadline, please inform us at rjsproduction@springernature.com immediately.

Please note that *Nature Methods* is a Transformative Journal (TJ). Authors may publish their research with us through the traditional subscription access route or make their paper immediately open access through payment of an article-processing charge (APC). Authors will not be required to make a final decision about access to their article until it has been accepted. [Find out more about Transformative Journals](https://www.springernature.com/gp/open-research/transformative-journals)

Your paper will now be copyedited to ensure that it conforms to Nature Methods style. Once proofs are generated, they will be sent to you electronically and you will be asked to send a corrected version within 24 hours. It is extremely important that you let us know now whether you will be difficult to contact over the next month. If this is the case, we ask that you send us the contact information (email, phone and fax) of someone who will be able to check the proofs and deal with any last-minute problems.

If, when you receive your proof, you cannot meet the deadline, please inform us at rjsproduction@springernature.com immediately.

Once your manuscript is typeset and you have completed the appropriate grant of rights, you will receive a link to your electronic proof via email with a request to make any corrections within 48 hours. If, when you receive your proof, you cannot meet this deadline, please inform us at rjsproduction@springernature.com immediately.

Once your paper has been scheduled for online publication, the Nature press office will be in touch to confirm the details.

Once your paper has been scheduled for online publication, the Nature press office will be in touch to confirm the details.

Content is published online weekly on Mondays and Thursdays, and the embargo is set at 16:00 London time (GMT)/11:00 am US Eastern time (EST) on the day of publication. If you need to know the exact publication date or when the news embargo will be lifted, please contact our press office after you have submitted your proof corrections. Now is the time to inform your Public Relations or Press Office about your paper, as they might be interested in promoting its publication. This will allow them time to prepare an accurate and satisfactory press release. Include your manuscript tracking number NMETH-A50796C and the name of the journal, which they will need when they contact our office.

About one week before your paper is published online, we shall be distributing a press release to news organizations worldwide, which may include details of your work. We are happy for your institution or funding agency to prepare its own press release, but it must mention the embargo date and Nature Methods. Our Press Office will contact you closer to the time of publication, but if you or your Press Office have any inquiries in the meantime, please contact press@nature.com.

Nature Portfolio journals [encourage authors to share their step-by-step experimental protocols](https://www.nature.com/nature-research/editorial-policies/reporting-standards#protocols) on a protocol sharing platform of their choice. Nature Portfolio 's Protocol Exchange is a free-to-use and open resource for protocols; protocols deposited in Protocol Exchange are citable and can be linked from the published article. More details can found at www.nature.com/protocolexchange/about.

Best regards,
Madhura

Madhura Mukhopadhyay, PhD
Senior Editor
Nature Methods